# High interannual surface $p\mathrm{CO_2}$ variability in the Southern Canadian Arctic Archipelago's Kitikmeot Sea.

Richard P. Sims[1], Mohamed M M Ahmed[1,2,3], Brian J. Butterworth[4,5], Patrick J. Duke[6], Stephen F. Gonski[7], Samantha F. Jones[1], Kristina A. Brown[8,9], Christopher J. Mundy[9], William J. Williams[8], Brent. G. T. Else[1],

[1]Department of Geography, University of Calgary, Calgary, Alberta, T2N 1N4, Canada
[2]Geology Department, Beni-Suef University, 101 Salah Salem St., Bani Sweif, 62511, Egypt
[3]Education and Research Group, Esri Canada, Calgary, Alberta, T2P 3T7, Canada
[4]Cooperative Institute for Research in Environmental Sciences, University of Colorado, Boulder, Colorado, USA
[5]NOAA Physical Sciences Laboratory, Boulder, Colorado, USA
[6]School of Earth and Ocean Sciences, University of Victoria, Victoria, British Columbia, Canada
[7]School of Marine Science and Policy, University of Delaware, Lewes, Delaware, USA
[8]Institute of Ocean Scieneces, Fisheries and Oceans Canada, Sidney, British Columbia, Canada
[9]Centre for Earth Observation Science, Depertment of Environment and Geography, University of Manitoba, Winnipeg, MB R3T 2N2, Canada

*Correspondence to*: Richard P. Sims (richardpeter.sims@ucalgary.ca)

**Abstract**. Warming of the Arctic due to climate change means the Arctic Ocean is now ice-free for longer, as sea ice melts earlier and refreezes later. Yet, it remains unclear how this extended ice-free period will impact carbon dioxide ($\mathrm{CO_2}$) fluxes due to scarcity of surface ocean $\mathrm{CO_2}$ measurements. Baseline measurements are urgently needed to understand spatial and temporal air−sea $\mathrm{CO_2}$ flux variability in a changing Arctic Ocean. There is also uncertainty as to whether the previous basin-wide surveys are representative of the many smaller bays and inlets that make up the Canadian Arctic Archipelago (CAA). By using a research vessel that is based in the remote Inuit community of Ikaluqtuutiak (Cambridge Bay , Nunavut), we have been able to reliably survey $p\mathrm{CO_2}$ shortly after ice melt and access previously unsampled bays and inlets in the nearby region. Here we present four years of consecutive summertime $p\mathrm{CO_2}$ measurements collected in the Kitikmeot Sea in the southern CAA. Overall, we found that this region is a sink for atmospheric $\mathrm{CO_2}$ in August (average of all calculated fluxes over the four cruises was -4.64 mmol m$^{-2}$ d$^{-1}$ ) but the magnitude of this sink varies substantially between years and locations (average calculated fluxes of +3.58, -2.96, -16.79 and -0.57 mmol m$^{-2}$ d$^{-1}$ during the 2016, 2017, 2018 and 2019 cruises, respectively). Surface ocean $p\mathrm{CO_2}$ varied by up to 156 µatm between years; highlighting the importance of repeat observations in this region, as this high interannual variability would not have been captured by sparse and infrequent measurements. We find that the surface ocean $p\mathrm{CO_2}$ value at the time of ice melt is extremely important in constraining the magnitude of the air−sea $\mathrm{CO_2}$ flux throughout the ice-free season. However, further constraining the air−sea $\mathrm{CO_2}$ flux in the Kitikmeot Sea will require a better understanding of how $p\mathrm{CO_2}$ changes outside of the summer season. Surface ocean $p\mathrm{CO_2}$ measurements made in small bays and inlets of the Kitikmeot Sea were ~20−40 µatm lower than in the main channels. Surface ocean $p\mathrm{CO_2}$ measurements made close in time to ice breakup (i.e., within 2 weeks) were ~50 µatm lower than measurements made >4 weeks after breakup. As previous basin-wide surveys of the CAA have focused on the deep shipping

channels and rarely measure close to the ice break-up date, we hypothesize that there may be an observational bias in previous studies, leading to an underestimate of the $CO_2$ sink in the CAA. These high-resolution measurements constitute an important new baseline for gaining a better understanding of the role this region plays in the uptake of atmospheric $CO_2$.

## 1 Introduction

The Arctic Ocean plays an important role in the global carbon cycle as a sink for atmospheric carbon dioxide ($CO_2$) (Bates and Mathis, 2009). Gas exchange and $CO_2$ drawdown is enhanced in cold polar surface waters because the solubility of $CO_2$ increases at low temperatures; this is known as the ocean solubility pump (Parmentier et al., 2013). Despite its role as a sink for $CO_2$, the magnitude of $CO_2$ uptake by the Arctic Ocean is poorly constrained as the region remains spatially and temporally under-sampled due to difficult seasonal access heavily skewing measurements to the ice-free summer period

(DeGrandpre et al., 2020). Additionally, logistical constraints in poorly charted nearshore waters also tend to bias underway $CO_2$ measurements to established shipping routes and the deep ocean basins, leaving much of the Arctic coastal zone under-sampled in the Surface Ocean $CO_2$ Atlas (SOCAT v2022) (Bakker, 2016).  This is not a trivial oversight, given that the Arctic Ocean is encircled by coasts and their associated shelf seas, 53% of the $\sim10.7\times10^6$ $km^2$ Arctic Ocean surface area is < 200m deep (Jakobsson, 2002).


The Arctic is already being heavily impacted by climate change (Landrum and Holland, 2020), with potentially devastating impacts on the Inuit and other Indigenous communities who live there (Ford et al., 2008). It is not certain how the Arctic carbon system will respond to climate change and how the effects of processes like ocean acidification will manifest and impact Inuit communities. Projecting long-term change in regions with complex biogeochemistry (i.e., the coastal domain) is

particularly difficult. To better predict how the Arctic carbon system will change in the future requires baseline measurements, including detailed surveys and regular monitoring of oceanic $p$CO$_2$, that reflect the diverse nature of Arctic marine environments.

The Canadian Arctic Archipelago (CAA) is made up of numerous islands that cover 13% of the Arctic Ocean (Macdonald et

al., 2010) and account for the bulk of Canada's 162,000 km of Arctic coastline (Wynja et al., 2015). The islands of the CAA form a complex bathymetry which is important in determining the circulation in the CAA (Wang et al., 2012). The majority of existing $p$CO$_2$ measurements made in the CAA were collected along the southern route through the Northwest Passage on the research icebreaker *CGGS Amundsen* (Ahmed et al., 2019). This large $p$CO$_2$ dataset was used to estimate a $-7.7 \pm 4$ Tg C yr$^{-1}$ sink for the CAA during the open water season (Ahmed and Else, 2019). The *CCGS Amundsen* $p$CO$_2$ dataset provides

excellent broad spatial coverage of the CAA, but the vast area surveyed was limited in temporal coverage and fine spatial detail. The *CCGS Amundsen* typically only transited through the central straits, channels, gulfs, and seas of the southern Northwest Passage once each summer. The numerous bays and inlets that are off the main channel were not sampled,

meaning that local-scale $pCO_2$ variability was potentially unaccounted for during the synoptic scale sampling. This small-scale $pCO_2$ variability is difficult to predict empirically and may be better observed via regional studies. For example, the

model of Ahmed et al. (2019) was shown to underestimate $pCO_2$ by an average of ~26 μatm in Coronation Gulf and Dease Strait regions of the Kitikmeot Sea. Ahmed et al. (2019) postulated that large river inflow in the region may account for divergences from their model, understanding whether this is the case warrants further investigation and makes the Kitikmeot Sea a prime location for focused study.

Our understanding of the inorganic carbon system in the Kitikmeot Sea region primarily comes from three distinct sources of measurements. Firstly, the 2010−2016 summertime ship measurements of $pCO_2$ in the central channel of the Kitikmeot presented by Ahmed et al. (2019). Their measurements show the region to be slightly undersaturated at the beginning of August, becoming slightly supersaturated in the middle of August through to the middle of September, and then becoming undersaturated again in early October. Coronation Gulf is one of the few areas of the CAA that was consistently observed to

be supersaturated with $CO_2$ in summer. Supersaturation of $pCO_2$ in Coronation Gulf is likely a result of high summer surface seawater temperatures ($CO_2$ thermodynamics mean that a 1°C temperature increase, increases $pCO_2$ by 4.23% (Takahashi et al., 1993)) and high river discharge, particularly to the southwest (Geilfus et al., 2018). The second source of carbonate system measurements in the region are $CO_2$ flux observations at the Qikirtaarjuk Island observatory on the Finlayson Islands in Dease Strait (Butterworth and Else, 2018). Their measurements from the 2017 ice breakup season

through to the summer indicate that there is $CO_2$ drawdown, and thus, undersaturation at breakup and for the first two weeks of open water. Near the end of July, the region transitions into a $CO_2$ source through to the end of August (Butterworth and Else, 2018). The region reverts to a sink in late August as the sea cools and surface $pCO_2$ declines; the region remains a sink until almost full ice cover in November (Butterworth et al., 2023 in preperation). A similar pattern was observed in the summer of 2018, except notably, when $pCO_2$ began to fall in late August the region did not revert all the way back into a

sink (Butterworth et al., 2023 in preperation). The third source of carbonate system measurements are provided by Duke et al. (2021) who report autonomous $pCO_2$ measurements at a depth of 7 m from an instrument installed on the Ocean Networks Canada (ONC) underwater sensor mooring in Cambridge Bay between August 2015 and August 2018. The sensor measurements from Cambridge Bay indicate that $pCO_2$ is supersaturated in winter and undersaturated by the start of June at the onset of sea ice melt (Duke et al., 2021). Their measurements show that there is a short period of supersaturation in the

middle of August coinciding with increased sea water temperature, the ocean then quickly returns to a $CO_2$ sink and remains undersaturated up until freeze-up (Duke et al., 2021). Duke et al. (2021) confirmed that the biogeochemical measurements at the ONC site were representative of the offshore during most seasons by comparing discrete dissolved inorganic carbon (DIC) and total alkalinity (TA) samples collected at both 2 and 7 m at the ONC platform and an offshore station (B1). The surface stratification at ONC breaks down after the 2 week sea ice melt and river runoff period in early July. After the sea ice

melt and river runoff period, DIC, TA, salinity, and temperature values recoreded by the ONC mooring are then once again representative of the surface mixed layer.

All three sources of measurements indicate that there is notable interannual variability in surface $pCO_2$ in the Kitikmeot Sea. The ship-based measurements provide a snapshot of spatial variability across the wider region during the open-water season whereas the time series from Qikirtaarjuk Island observatory and the ONC mooring provide insights into seasonal and interannual variability at specific locations. There are obvious shortcomings to both approaches. Icebreaker-based studies may under-represent small-scale variability that exists in nearshore regions that are inaccessible due to the vessel's large draft. Whereas the fixed observatories may over-represent temporal variability which is location-specific; for example, the ONC mooring is in an enclosed Bay close to the outlet of a river (Manning et al., 2020) and the flux footprint of the to Island observatory spans a hotspot for mixing and productivity (Dalman et al., 2019). Given the limitations of each of these data sources, there is a need to understand how representative they are of the wider Kitikmeot Sea region.

In this paper, we present surface $pCO_2$ measurements made during annual summertime surveys of the Kitikmeot Sea between 2016 and 2019. We use these new $pCO_2$ measurements to determine the magnitude of $CO_2$ uptake in the Kitikmeot Sea shortly after ice breakup. These new $pCO_2$ measurements allow us to bridge the gap between previous measurements, which were made at contrasting spatial scales (e.g., the low spatial variability point-scale observation from the local carbon observatories and the large-scale CAA-wide $pCO_2$ measurements),. We use our new measurements to explore whether there are small-scale regional $pCO_2$ differences in the inlets and bays of the CAA which are not adequately represented by CAA-wide sampling. We also use our new measurements to explore $pCO_2$ variability in the proximity of these observatories to determine whether they are representative of the wider region. In attempting to unify existing measurements, we aim to unravel the seasonal and interannual variability of $pCO_2$ in the region.

## 2 Methods

### 2.1 Oceanographic setting

The Kitikmeot Sea (Figure 1) is a shallow shelf sea within the CAA that encompasses Coronation Gulf to the west, linked via Dease Strait to Queen Maud Gulf in the East, Bathurst Inlet to the South, and Chantrey Inlet to the Southeast (Williams et al., 2018). The communities of Cambridge Bay, Kugluktuk, and Gjoa Haven, Nunavut, are the main year-round settlements in the Kitikmeot Sea region. River inputs from mainland Canada and snow and ice melt provide a considerable source of freshwater in the region (Williams et al., 2018), resulting in some of the lowest salinity surface waters in the CAA (Ahmed et al., 2019). The Kitikmeot sea is strongly nitrogen limited (Back et al., 2021) with surface nitrate concentrations of 1.3 μmol $L^{-1}$ (Dalman et al., 2019), and as a result chlorophyll concentrations are also low in the region (Kim et al., 2020). Observations and modelling of the physical oceanography of the region demonstrates that the stratification regime in Dease Strait and Queen Maud Gulf is characterised by a ~40 m warm fresh surface layer and a cold salty bottom layer which extends down to around 100 m (Xu et al., 2021). Coronation Gulf has a three layer regime composed of a 40 m warm fresh

surface layer, a colder salty layer down to 100 m and a stable deep layer down to 350 m (Xu et al., 2021). Vertical mixing in the Kitikmeot Sea is prohibited by strong stratification throughout most of the year; however after sea ice breakup wind driven mixing gradually deepens the surface mixed layer resulting in an almost fully mixed water column in Dease Strait (Xu et al., 2021).

The oceanographic boundary for the Kitikmeot Sea has been designated as where the shelf shoals to <30 m in the west (Dolphin and Union Strait) and northeast (Victoria Strait) (Williams et al., 2018).  At the Dolphin and Union Strait, warm fresh surface seawater flows out across the sills and subsurface flows of more saline nutrient-rich Pacific waters enter the sea. Another feature of the Kitikmeot Sea is that strong tidal currents in narrow channels can keep certain areas ice-free in winter (Williams et al., 2018). Strong tidal currents beneath sea ice such as around the Finlayson Islands in Dease Strait act to slow winter sea ice growth and enhance primary production by introducing nutrients (Dalman et al., 2019). First-year sea ice dominates the Kitikmeot Sea although some multiyear ice may be blown into Queen Maud Gulf from the northern part of the CAA (Xu et al., 2021). Seawater temperatures across the Kitikmeot Sea vary considerably throughout the year; they are around -2°C in winter and reach upwards of 10°C in summer (Xu et al., 2021). The bounding sills, large freshwater inputs and low nutrient loads make the Kitikmeot Sea unique within the CAA.

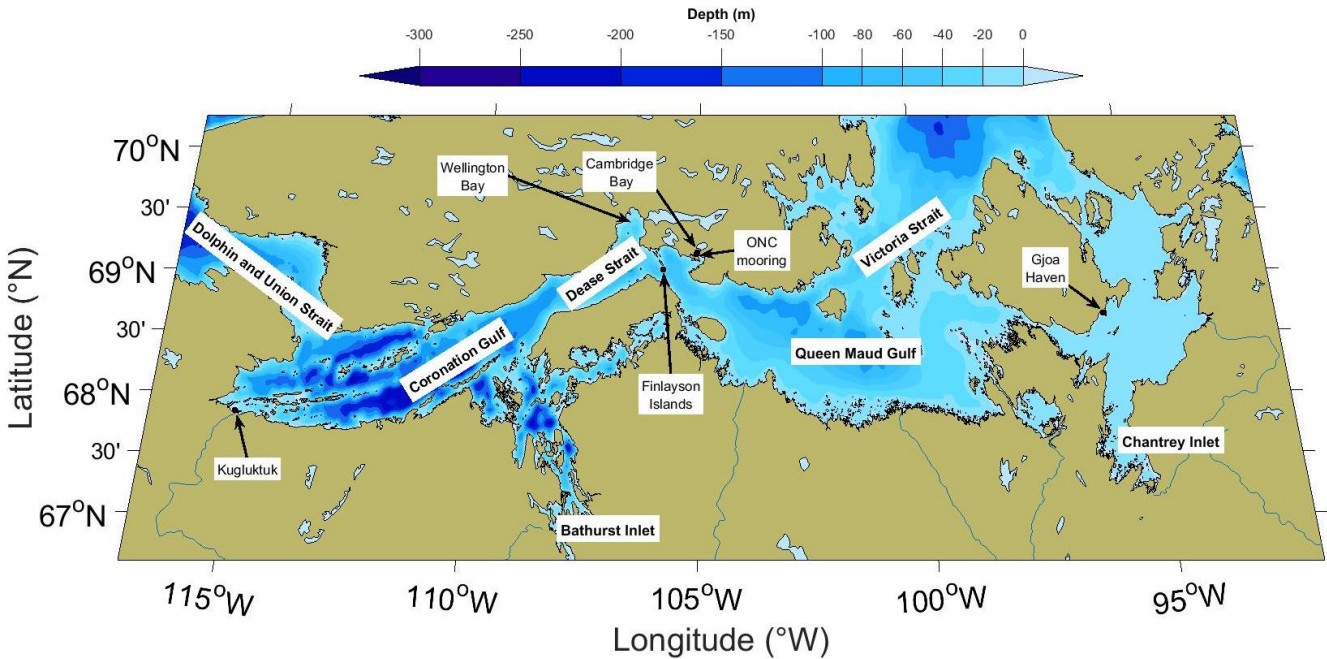

Figure 1: A map of the Kitikmeot Sea. The main settlements in the region (Cambridge Bay, Kugluktuk and Gjoa Haven) are labelled as are the Ocean Networks Canada mooring and the Finlayson Islands (where the Qikirtaarjuk Island observatory is located). Shoreline data was taken from the World Vector Shoreline database and river data was taken from the CIA World Data Bank II (WDBII), both of which were accessed via the Global Self-consistent, Hierarchical, High-resolution Geography Database (GSHHG) (Wessel and Smith, 1996). Bathymetry data was taken from the 2-minute Gridded Global Relief Data (ETOPO2) v2 database (NGDC, 2006). This map was made using tools from the M_Map Matlab plotting package (Pawlowicz, 2020).

## 2.2 Field campaign description

Annual oceanographic surveys of the summertime surface seawater partial pressure of carbon dioxide ($p$CO$_2$ $_{(sw)}$) were conducted between 2016 and 2019 in the Kitikmeot Sea (Figure 1) aboard the *RV Martin Bergmann* as part of the Marine Environmental Observation, Prediction and Response Network (MEOPAR) and Kitikmeot Sea Science Study (K3S) programs (cruise details in table S1). In each of the four years, an underway $p$CO$_2$ system was deployed on cruises conducted under ice-free conditions between early August and mid-September. The Canadian High Arctic Research Station (CHARS) in Cambridge Bay, Nunavut acted as a staging ground for this work as Cambridge Bay is the home port for the *RV Martin Bergmann*.

Between 2016 and 2019, the cruise track varied from year to year depending on the objectives of the research conducted (Figure 2). The first week of each summer field season was typically used to complete work for the MEOPAR program, the majority of the ship time for the MEOPAR work was spent in the proximity of Cambridge Bay, the Finlayson Islands, Wellington Bay and the western region of Queen Maud Gulf. Cruises in mid to late August were used to conduct work for the K3S program; for the K3S work the ship typically travelled further from Cambridge Bay heading into Bathurst Inlet, the

central region of Queen Maud Gulf and Chantrey Inlet. The opportunistic nature of the data collection meant that data
density varied between regions, as not every region was surveyed each year.

Sea ice concentrations in the months preceding each annual survey were taken from the daily gridded 3.125 km AMSR2
satellite radiometer product (Spreen et al., 2008). To determine weeks since open water, the nearest point on the AMSR2
grid was determined for each $pCO_{2\,(sw)}$ measurement. The time between the measurement and when sea ice concentration fell
constantly below the threshold value for the marginal ice zone (85%) (Cruz-García et al., 2021) was then calculated.

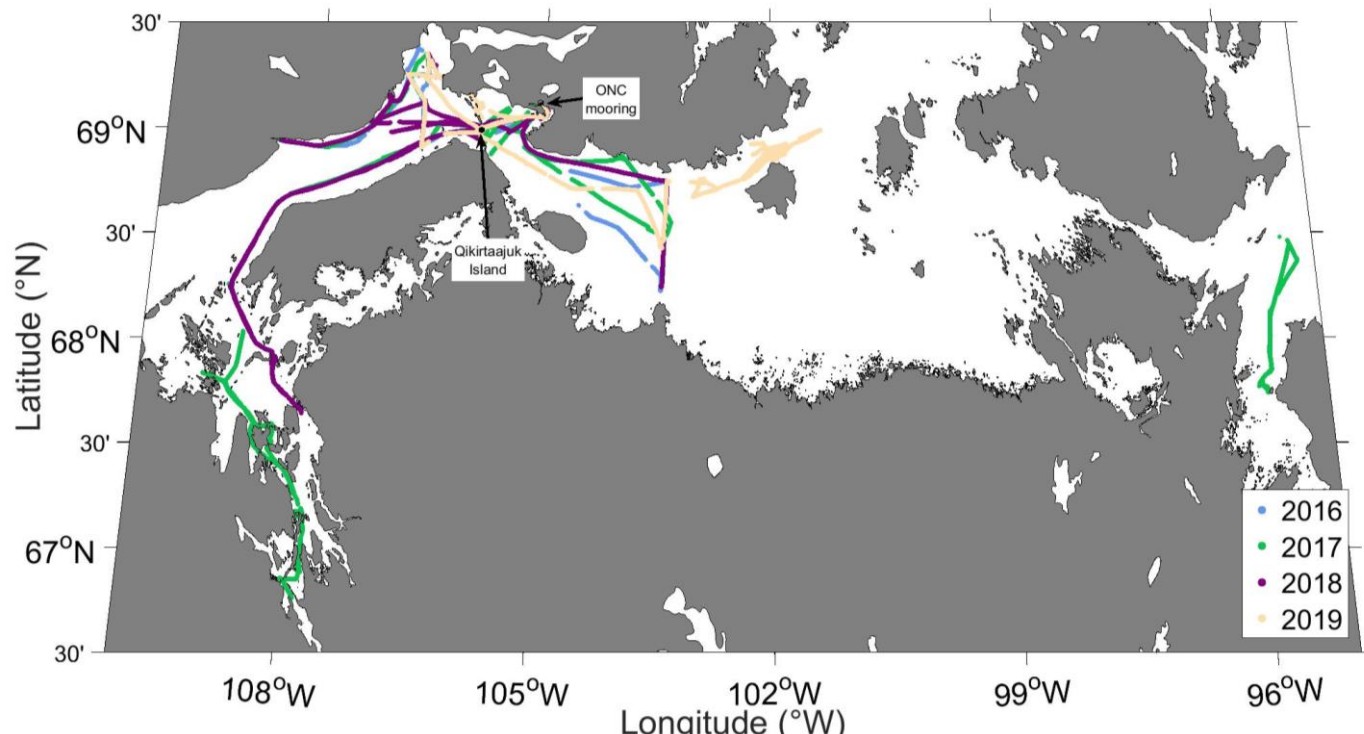

**Figure 2: Ship cruise tracks for each of the four surveyed years. The Ocean Networks Canada mooring and the Qikirtaarjuk Island Observatory where the eddy covariance tower is located are shown by black dots.**

**2.3 Underway system**

The *RV Martin Bergmann* is a 20 m repurposed commercial fishing trawler from Newfoundland with a draft of 3.4 m
(Figure 3a and 3b). The ship does not have its own dedicated integrated underway system; instead surface seawater was
sampled from an inlet at a depth of ~1 m through ~2 m of 1/2" ID PVC tubing securely draped over the bulwark of the vessel
through an external hatch (Figure 3c and 3d). A Waterra Tempest WSP-12V-3 submersible pump was used to pump surface
seawater through this inlet tubing at a rate of 10 L min$^{-1}$. *In situ* surface seawater temperature (SST$_{(1m)}$) was measured by a
Campbell Scientific 107 temperature sensor (error of $<\pm0.01°C$ over the measurement range) attached to the tubing inlet.

Upon entering the ship, the flow of seawater passed through a SoMAS MSRC VDB-1 vortex debubbler and was split between several instruments via Tygon tubing (Figure 3). An Idronaut Ocean Seven 315 On-line module thermosalinograph measured seawater temperature ($SST_{(tsg)}$) with an accuracy of 0.003 °C and conductivity with an accuracy of 0.003 mS cm$^{-1}$ at a seawater flowrate of 0.5 L min$^{-1}$. A Wetlabs ECO BBFL2B Triplet measured fluorescence with a sensitivity of 0.025 µg/L at a flowrate of 2.5 L min$^{-1}$. The output of the ECO fluorescence sensor was post-processed to remove spikes from bubbles and particles but was not calibrated against *in situ* measurements. A flow of 2 L min$^{-1}$ was directed to the seawater equilibrator. Instrument flowrates were set with manual flowmeters so that the internal instrument volumes and associated tubing of the Idronaut, ECO and equilibrator were flushed at the same rate, this meant that approximately half of the 10 L min$^{-1}$ flow from the pump was not analysed and was discarded overboard.

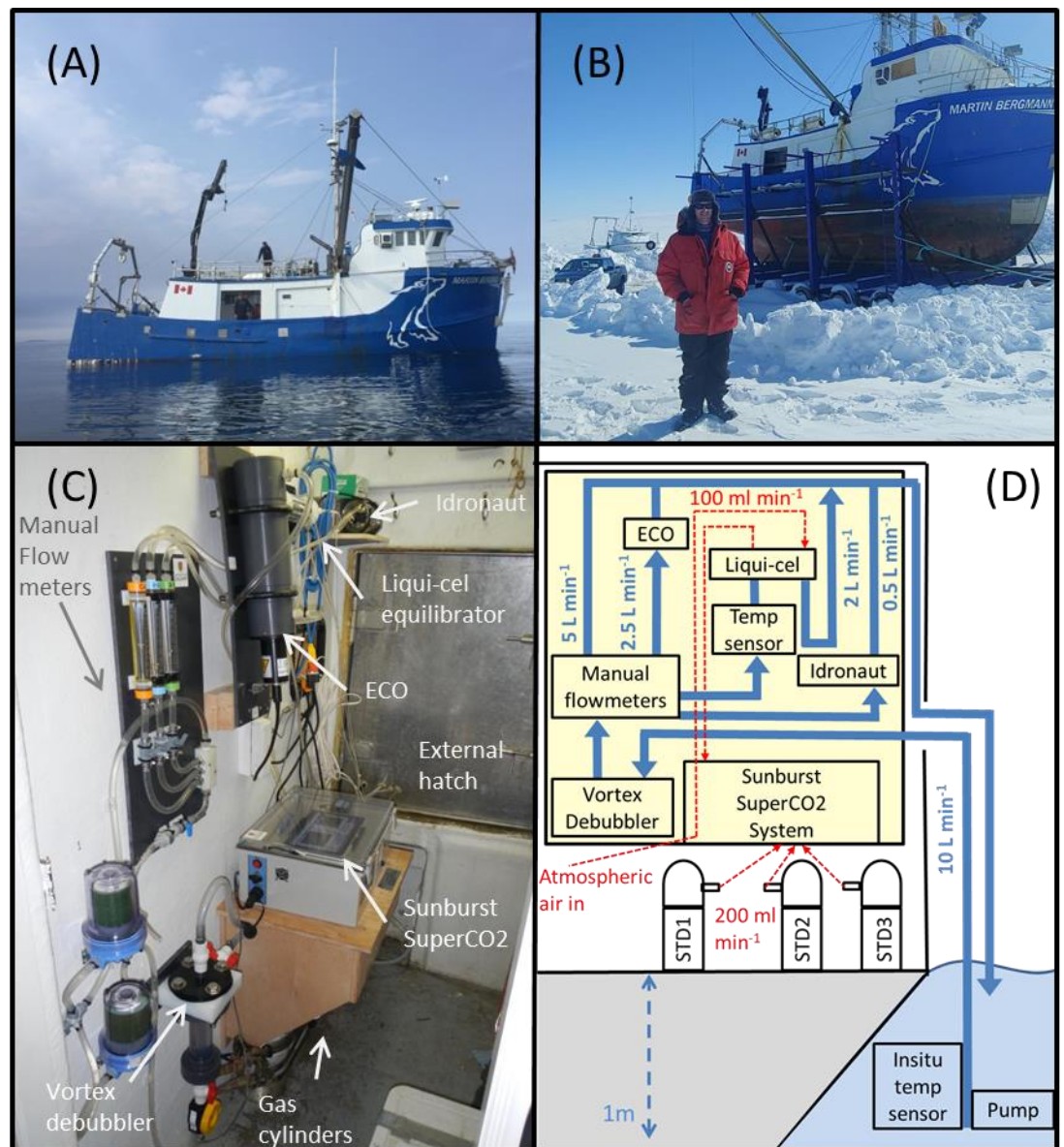

**Figure 3: (a) Image of the *RV Martin Bergmann* at sea taken in August 2017, (b) image of *RV Martin Bergmann* stored on its trailer taken on a mild day in May 2019, (c) labelled photograph of the underway system installed in the ship's lab space, and (d) detailed cross sectional schematic of the underway system with labelled instruments and flowrates. Instruments mounted to the wall are shown with a yellow background, water circulation is shown in blue and air circulation is shown in red.**

A commercially available Sunburst Sensors underway SuperCO$_2$ system measured surface seawater CO$_2$; an identical system was previously described by Evans et al. (2019). The SuperCO$_2$ system follows the general recommendations of Dickson et al. (2007) SOP5. A Permapure liqui-cel 2.5X8 series membrane contactor was used as the equilibrator for the $p$CO$_2$ system,

the waterside seawater flowrate for the equilibrator was approximately 2 L min⁻¹. Seawater temperature was measured at the equilibrator seawater inlet using a thermistor ($T_{(equ)}$). The gas counter flow into the equilibrator was supplied by an air pump at a flowrate of 100 ml min⁻¹. $CO_2$ has been shown to fully equilibrate in this model liqui-cel when set up in a single pass setup at these water and gas flowrates (Sims et al., 2017). The system does not utilise a dryer and thus does not require a water vapour correction in post-processing as the equilibrator is assumed to be at 100 % humidity. For additional accuracy, the inbuilt $H_2O$ sensor was calibrated with a LI-610 Portable Dew Point Generator on-site before each deployment, the dew point generator has an accuracy of ± 0.2 °C. The SuperCO$_2$ system has a standard multi-position valve and alternates between equilibrator air, atmospheric samples, and three gas standards. The timing of the valve switching was set so that each of the three $CO_2$ standards ($CO_2$ mixing ratios ($xCO_2$) of 255.1, 409.9, and 566.4) were flushed through the system at 200 ml min⁻¹ for 5 minutes every 6 hours. Standard gases were certified at the University of Manitoba against standards obtained from Environment and Climate Change Canada, and are thus traceable to World Meteorological Organization standards. The SuperCO$_2$ system has an integrated air pump configured to make atmospheric measurements; these measurements were not used due to contamination from the ship's exhaust. The SuperCO$_2$ system also measured atmospheric pressure $P_{(atm)}$.

Measurements from the underway system were logged every minute. $xCO_2$ and related variables were logged to the computer of the SuperCO$_2$ system, the data recorded by the ECO were logged to a separate data file, and the latitude and longitude recorded with a Garmin GPS16X-HVS GPS unit were logged to a Campbell Scientific CR300 data logger. The $CO_2$ measured by the system were processed following SOP 5 (Dickson et al., 2007). $xCO_2$ is the output provided by the Licor 850 in the SuperCO$_2$ system. $xCO_2$ is calibrated using a piecewise linear interpolation in time with the three standards. As there was no dryer the equilibrator is assumed to be at full humidity, the partial pressure in the equilibrator ($pCO_{2\,(equ)}$) was therefore calculated by multiplying by atmospheric pressure $P_{(atm)}$. $pCO_{2\,(equ)}$ was converted to $pCO_{2\,(1m)}$ using the $T_{(equ)}$, $SST_{(1m)}$, and the fractional temperature change constant of Takahashi et al. (1993). The depth of the seawater inlet was validated each year by comparing the thermosalinograph salinity and the in situ temperature sensor with surface temperature and salinity from CTD rosette measurements at the surface. As there was no in situ temperature sensor during the 2017 and 2018 field seasons, the warming was characterised from $T_{(equ)}$ and CTD rosette measurements following Ahmed et al. (2019), details of this can be found in the supplementary materials. Additionally, median observational values of -0.17°C and +0.1 were added to the in situ temperature and salinity to account for ubiquitous skin effects when calculating interfacial seawater $pCO_2$ (Woolf et al., 2019).

Using an identical setup, DeGrandpre et al. (2020) estimate the $pCO_2$ uncertainty as ± 5 μatm, this is the uncertainty for our 2016 and 2019 measurements. In 2017 and 2018, there is an additional uncertainty component associated with using an empirical relationship to obtain $SST_{(1m)}$. This additional uncertainty was calculated by taking the Root-mean-square deviation(RMSD) values from those empirical relationships (2017 = 0.49°C, 2018 = 0.64°C) and propagating them through

the temperature equation for $pCO_{2\,(1m)}$ (Takahashi et al., 1993). This resulted in an additional 2.09% and 2.74% uncertainty in $pCO_{2\,(1m)}$, these values are similar to the 2% uncertainty reported by Ahmed et al. (2019) following the same method. For a $pCO_{2\,(equ)}$ value of 300 µatm this equates to an additional 6.3 and 8.2 µatm uncertainty for each year respectively. Propagating uncertainties gives average uncertainties of 8.04 and 9.60 µatm for 2017 and 2018 respectively. The calculation of the 2017 and 2018 uncertainties is consistent with the International Bureau of Weights and Measures (BIPM) Guide to the expression of uncertainty in measurement (GUM) methodology (JCGM, 2008).

The standard system configuration during the four cruises is detailed above; changes from this configuration during specific cruises are detailed in the supplementary materials (Table S2). There are several logistical aspects associated with deploying, operating, and maintaining an underway $pCO_2$ system in a remote Arctic location on a small vessel like the *RV Martin Bergmann;* this is discussed further in supplementary materials.

## 2.4 Calculations: Air−sea CO$_2$ fluxes

In the absence of a reliable ship-based atmospheric $CO_2$ record, hourly measurements were taken from the atmospheric observatory in Barrow Alaska (71.32°N,156.61°W) (K.W. Thoning, 2020;Peterson et al., 1987). Despite the long distance between Barrow and the Kitikmeot Sea (around 1800 km), atmospheric $CO_2$ are very similar at both locations as the atmosphere is well mixed for a long residence time gas like $CO_2$ and both locations are remote northern sites away from biogenic and industrial emissions. To validate this assumption a long term (1985−2019) mean difference of 0.246 µatm was calculated between the hourly measurements at Barrow and weekly atmospheric samples from Alert Nunavut (Lan et al., 2022). Wind speed adjusted to a reference height of 10 m ($U_{10}$) was taken from the Qikirtaarjuk Island observatory (Butterworth and Else, 2018) for the 2017 and 2018 field seasons whereas a four times daily record of $U_{10}$ from the NCEP-DOE v2 reanalysis product (Kalnay et al., 1996) was used for 2016 and 2019 field seasons.

The air–sea fluxes of $CO_2$ (F, mmol m$^{-2}$ d$^{-1}$) was calculated as

$$F_{(sea-air)} = k_W\, k_0\, \Delta pCO_2$$

The water phase gas transfer velocity ($k_w$, cm hr$^{-1}$) was calculated using $U_{10}$ and the parameterisation of Nightingale et al. (2000), a unitless Schmidt number (Sc) normalised to a Sc of 660 (Wanninkhof, 2014) was used to scale $k_w$.

$$k_W = (0.222\,(U_{10})^2 + 0.333\,(U_{10}))\,(Sc/660)^{-1/2}$$

$\Delta pCO_2$ (µatm) is the partial pressure difference between the seawater interface and air $\Delta pCO_2 = pCO2_{(sw)} - pCO2_{(air)}$. The solubility of $CO_2$ in seawater ($k_{0,}$ mol L$^{-1}$ atm$^{-1}$) was taken from Weiss (1974). The Schmidt number and solubility were calculated using the *in situ* temperature and salinity values adjusted for skin effects (Woolf et al., 2019).

Direct measurements of the air–sea $CO_2$ fluxes (F $_{(sea-air)}$) made using the micrometeorological eddy covariance technique (Butterworth and Else, 2018) were used to infer $pCO_{2(sw)}$ by rearranging the flux equation. That was achieved using $pCO_{2(air)}$ from the Licor 7200 at the Qikirtaarjuk Island observatory and SST and SSS from a mooring at a depth of 13 m which was 1

km from the eddy covariance tower (Butterworth et al., 2023 in preperation). An eddy covariance flux footprint is the area over which the eddy covariance measurements correspond to and varies depending on atmospheric conditions. Using the Kljun et al. (2015) footprint model, Butterworth and Else (2018) showed that the footprint of the Qikirtaarjuk Island observatory during spring and summer can be modelled as an ellipse with an upwind axis that varies between approximately 0.75 – 2.0 km and a cross-wind axis that varies between 0.1 – 0.2 km. The effective flux footprint is however much smaller as over 90% of the flux signal comes from within 100 m of the eddy covariance tower. Uncertainty in the $pCO_{2(sw)}$ values derived using eddy covariance arises from uncertainty in the flux measurements (hourly uncertainty of ~20% in the Arctic) (Dong et al., 2021a), uncertainty in the gas transfer parameterisation (~ 5–10%) (Woolf et al., 2019), the small uncertainty in the atmospheric $pCO_2$ value, uncertainties in $k_0$ and the schmidt number (including uncertainties in SST and salinity inputs from the 13 m mooring).

$$(F_{(sea\text{-}air)} / k_W\, k_0\,) + pCO_{2(air)} = pCO_{2(sw)}$$

## 3. Results

To facilitate comparisons between the four summertime cruises, observations have been partitioned into separate oceanographic zones based on the local geography, observational data density, previous $pCO_{2\,(sw)}$ measurements, and proximity to the local carbon observatories (Figure 4a). Bathurst Inlet and Chantrey Inlet were designated zones based on their large freshwater inputs. The Finlayson Islands and Cambridge Bay are where the Qikirtaarjuk Island observatory and ONC mooring are located, respectively; these regions were also heavily surveyed because the *RV Martin Bergmann* often returned to port in Cambridge Bay and passed the islands to access Wellington Bay and Bathurst Inlet. Wellington Bay (Figure 1) is a shallow, partially enclosed basin for which a relatively large amount of data was collected due to annual fish-tagging surveys associated with the local subsistence char fishery (Harris et al., 2020). All the measurements in the Dease Strait West zone were made in the central channel and are in the same approximate geographical region to those collected by Ahmed et al. (2019). Most of the measurements in the Queen Maud Gulf zone were made in the west; the box is large enough to include sparse measurements in the central and Northern regions which do not warrant being considered separately.

Observations of temperature, salinity, $pCO_{2\,(sw)}$, fluorescence, $U_{10}$, and $CO_2$ flux during the four field seasons are plotted as time series and coloured by the sub-region of the measurement (Figure 4b-4g). Summary statistics (mean, standard deviation, and range) of each variable in each region for all four cruises are presented in Table 1. Plots showing the timing of the cruise track, temperature, salinity, $pCO_{2(sw)}$, and chlorophyll-a fluorescence can be found in the supplementary materials (Figures S2 to S6).

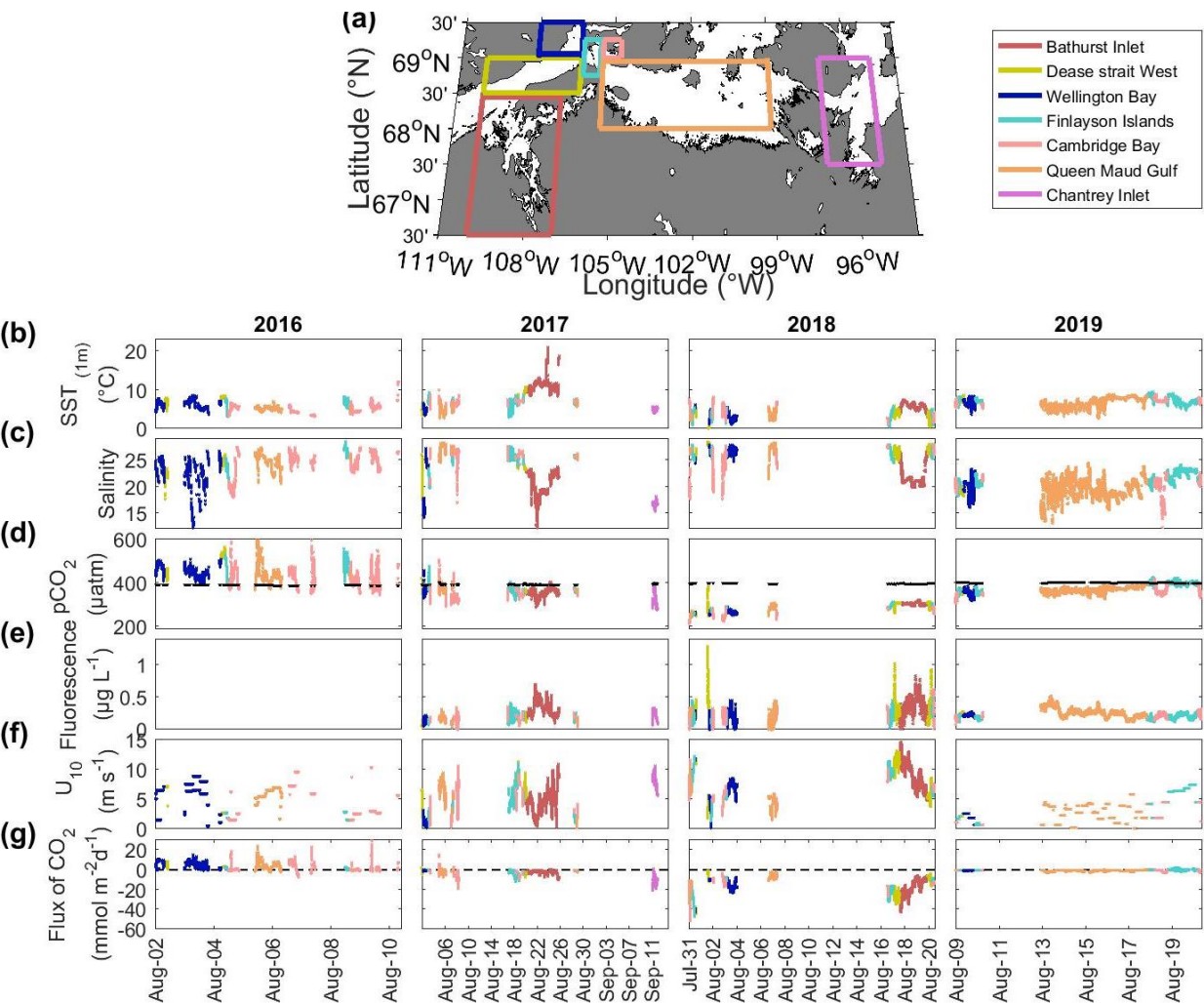

**Figure 4: (a)** Map of Kitikmeot Sea showing the region surveyed by the *RV Martin Bergmann* between 2016 and 2019. The sampled region was subdivided as described in the main text; these sub regions are shown as coloured boxes and correspond to the names in the legend. Timeseries subplots of underway surface ocean (1m) observations for 2016 through to 2019 of **(b)** SST (1m), **(c)** salinity, **(d)** $pCO_2$ (sw) (with $pCO_2$ (atm) in black), **(e)** fluorescence, **(f)** U10, and **(g)** flux of $CO_2$ (no flux is indicated by a dashed black line). The time series data are coloured according to the sampling regions in panel (a). The period of measurements was not consistent between years so the date label tick spacing and the range are different between years. Large data gaps correspond to when the ship was in port between cruise legs or data outages. An alternate version of this figure where the y-axes are not normalised between years is included in the supplement (Figure S7).

SST $_{(1m)}$ interannual variability was on the order of several degrees (Figure 4b), for example the SST $_{(1m)}$ was lowest in 2018 (4.3 °C) and highest in 2017 (8.4 °C) (Table 1). Inter-region SST $_{(1m)}$ differences of ~10°C were observed during all four surveys, for example in 2016 the range is SST was 3.18 – 12.13°C (Table 1). Summertime warming can be observed in the data for certain sub regions which were visited multiple times such as Cambridge Bay in 2016 (SST $_{(1m)}$ trend of +0.11°C d$^{-1}$

from 5 August 2016 to 10 August 2016) or were sampled for a continuous period such as Queen Maud Gulf in 2019 (SST $_{(1m)}$ trend of +0.64°C d$^{-1}$ from 13 August 2019 to 19 August 2019)  (Figure 4b). Some of the sub regions were considerably warmer than others (e.g., Bathurst Inlet was 2.82 °C warmer in 2017 and 1.51 °C warmer  in 2018 compared to the measurement averages for those respective years), whereas other regions were consistently colder (e.g.,. Queen Maud Gulf was 3.45 °C colder in 2017 and 0.76 °C colder compared to the measurement averages for those respective years)  (Table 1).


There was large interannual variability in surface salinity; for example average observed salinity in 2019 was 20.12 compared with 24.82 in 2018 (Figure 4c). Salinity values were much lower in Chantrey Inlet in 2017 (16.61) and Bathurst Inlet in both 2017 (20.78) and 2018 (21.86) relative to the salinities in other regions in those years (Table 1). Salinity ranges on the order of ~10 were observed between regions in all years, for example in 2018 the maximum salinity range was 10.84.

The salinity data are marked by rapid changes of ~5 which did not coincide with equivalent temperature changes (Figure 4c); these salinity transitions are evident in the 2017 and 2018 Bathurst Inlet data, much of the Cambridge Bay data and the Wellington Bay data from 2016 and 2019. There is evidence of freshening in Wellington Bay from 2 August 2016 to 4 August 2016 (salinity trend of -0.87 d$^{-1}$) and in Queen Maud Gulf in 2019 (salinity trend of 0.11 d$^{-1}$ from the 13 August 2019 to 19 August 2019), but there does not appear to be a seasonal freshening trend in 2017 or 2018.


There was high interannual $pCO_{2\,(sw)}$ variability (Table 1), average measured $pCO_{2\,(sw)}$ was supersaturated (445μatm) in 2016, undersaturated in 2017 (361 μatm) and 2019 (373 μatm) and highly undersaturated in  2018 (288 μatm) (Figure 4d). There was also high regional variability in   $pCO_{2\,(sw)}$ each year, for example in 2018 $pCO_{2\,(sw)}$ ranged from  218 μatm to 387 μatm (Table 1). There were identifiable trends in $pCO_{2\,(sw)}$ across all regions in 2018 and 2019 (Figure 4d); for example,

$pCO_{2\,(sw)}$ increased by 2.22 μatm d$^{-1}$ from 31 July 2018 to 21 August 2018 and 4.04 μatm d$^{-1}$ from the 9 August 2019 to 21 August 2019. In all four years, Cambridge Bay had lower $pCO_{2\,(sw)}$ relative to the other regions, for example the average $pCO_{2(sw)}$ in Cambridge Bay in 2019 was 359 μatm whereas the averages in the Finlayson Islands and Queen Maud Gulf were 392 μatm  and 370 μatm respectively (Table 1). Low $pCO_{2\,(sw)}$ values were also seen in Bathurst Inlet (e.g., 359 μatm in 2017), Chantrey Inlet (e.g., 326 μatm in 2017) and Wellington Bay (e.g., 268 μatm in 2018) (Table 1) . Many low $pCO_{2\,(sw)}$

regions were also low salinity regions, for example Chantrey Inlet and Wellington Bay in 2017 (Table 1). Fluorescence was generally low throughout all the cruises, in all years, except for the relatively higher fluorescence signal in Bathurst Inlet and around the Finlayson Islands (Figure 4e). The air–sea $CO_2$ flux (Figure 4g) reflects the trends in the predictor variables, particularly $pCO_{2\,(sw)}$ and $U_{10}$ (Figure 4d and 4f). The air–sea flux calculated in 2016 was 3.58 mmol m$^{-2}$ d$^{-1}$ reflecting the fact that the $pCO_{2\,(sw)}$ was supersaturated. In 2017 and 2019 surface ocean $pCO_{2\,(sw)}$ was quite undersaturated (361 and 373

μatm respectively), the 2017 flux was larger (-2.96 mmol m$^{-2}$ d$^{-1}$)  than the 2019 flux (-0.57 mmol m$^{-2}$ d$^{-1}$) as the wind speed was very low in 2019 (3.1 ms$^{-1}$). As $pCO_{2\,(sw)}$ was highly undersaturated (288 μatm) in 2018, there was a large flux into the ocean  -16.79 mmol m$^{-2}$ d$^{-1}$.

Table 1: Underway surface ocean (1m) observation summary table for the *RV Martin Bergmann* cruises from 2016 through 2019. Geographical sub regions are defined in Figure 4a. Top line is the mean ± 1 standard deviation and the bottom row is the measurement range. Table averages are the average of all the observations for each variable for each year and have not been scaled to the spatial extent of each region.

| Year | Sub region | No of obs | $SST_{(1m)}$ (°C) | Salinity | $pCO2_{(sw)}$ (µatm) | Fluorescence | $U_{10}$ (m s$^{-1}$) | Flux (mmol m$^{-2}$ d$^{-1}$) |
|---|---|---|---|---|---|---|---|---|
| 2016 | Dease Strait West | 376 | 7.62 ± 0.75 4.71 – 8.50 | 23.58 ± 1.54 17.74 – 26.08 | 490.66 ± 46.38 411.51 – 567.38 | - | 4.25 ± 2.31 1.12 – 7.22 | 4.37 ± 1.71 0.64 – 8.77 |
| | Wellington Bay | 1523 | 6.35 ± 1.10 3.68 – 8.66 | 22.37 ± 3.11 12.21 – 26.84 | 455.98 ± 26.26 393.24 – 510.08 | - | 6.05 ± 2.23 0.58 – 8.91 | 5.32 ± 3.50 -0.23 – 15.19 |
| | Finlayson Islands | 412 | 6.30 ± 1.46 3.32 – 8.25 | 25.67 ± 1.77 21.01 – 28.57 | 471.05 ± 40.86 383.13 – 560.18 | - | 2.19 ± 0.59 1.55 – 2.92 | 1.23 ± 0.99 -0.15 – 3.38 |
| | Cambridge Bay | 2051 | 5.18 ± 1.38 3.18 – 12.13 | 24.42 ± 2.29 18.06 – 27.51 | 423.87 ± 43.44 347.40 – 656.73 | - | 4.22 ± 2.69 1.50 – 10.37 | 2.24 ± 4.45 -7.23 – 32.88 |
| | Queen Maud Gulf | 1173 | 5.38 ± 0.56 4.22 – 7.14 | 24.61 ± 1.54 20.71 – 27.37 | 444.26 ± 57.26 372.35 – 749.17 | - | 5.93 ± 1.08 1.55 – 6.97 | 4.24 ± 4.00 -2.02 – 24.30 |
| | Average all | 5535 | 5.80 ± 1.34 3.18 – 12.13 | 23.93 ± 2.57 12.21 – 28.57 | 445.08 ± 47.37 347.40 – 749.17 | - | 4.94 ± 2.45 0.58 – 10.37 | 3.58 ± 4.08 -7.23 – 32.88 |
| 2017 | Bathurst Inlet | 7426 | 11.24 ± 1.90 8.56 – 21.14 | 20.78 ± 2.04 11.04 – 23.88 | 358.80 ± 16.47 291.75 – 407.48 | 0.32 ± 0.12 0.10 – 0.71 | 4.54 ± 2.22 0.38 – 10.91 | -2.11 ± 1.87 -10.32 – 0.70 |
| | Dease Strait West | 1137 | 8.27 ± 1.93 3.40 – 10.60 | 23.21 ± 1.59 14.63 – 26.04 | 367.92 ± 6.24 352.31 – 420.93 | 0.18 ± 0.07 0.04 – 0.29 | 6.89 ± 2.32 0.30 – 11.43 | -3.83 ± 2.24 -9.31 – 2.55 |
| | Wellington Bay | 847 | 5.04 ± 0.76 | 20.08 ± 4.60 | 361.68 ± 15.08 334.16 – | 0.14 ± 0.03 0.07 – 0.22 | 1.27 ± 0.60 0.29 – | -0.28 ± 0.24 -1.38 – 0.34 |

| Year | Location | N | | | | | | |
|---|---|---|---|---|---|---|---|---|
| | | | 3.55 – 7.20 | 14.23 – 27.22 | 459.24 | | 3.12 | |
| | Finlayson Islands | 3491 | 6.95 ± 0.83 3.08 – 9.39 | 25.18 ± 1.38 19.86 – 27.60 | 372.81 ± 15.10 324.66 – 478.13 | 0.20 ± 0.06 0.04 – 0.42 | 4.53 ± 2.31 0.43 – 11.12 | -2.10 ± 2.12 -11.51 – 0.70 |
| | Cambridge Bay | 1951 | 6.47 ± 0.73 3.63 – 9.99 | 26.14 ± 1.48 17.09 – 28.14 | 350.80 ± 27.19 294.77 – 506.91 | 0.15 ± 0.06 0.00 – 0.36 | 5.05 ± 2.46 0.43 – 10.69 | -4.08 ± 4.61 -18.76 – 15.80 |
| | Queen Maud Gulf | 1519 | 4.97 ± 1.23 2.78 – 7.50 | 27.25 ± 1.05 24.58 – 28.31 | 378.39 ± 11.69 346.81 – 422.62 | 0.17 ± 0.03 0.07 – 0.32 | 6.64 ± 2.28 0.29 – 9.46 | -1.97 ± 1.73 -7.11 – 1.76 |
| | Chantrey Inlet | 1102 | 4.97 ± 0.40 4.09 – 5.76 | 16.61 ± 0.57 15.45 – 18.25 | 325.91 ± 34.66 280.74 – 403.54 | 0.23 ± 0.06 0.07 – 0.34 | 8.33 ± 1.15 5.69 – 10.73 | -11.60 ± 4.49 -20.77 – 1.16 |
| | Average all | 17473 | 8.42 ± 2.95 2.78 – 21.14 | 22.68 ± 3.51 11.04 – 28.31 | 361.07 ± 22.20 280.74 – 506.91 | 0.24 ± 0.12 0.00 – 0.71 | 5.02 ± 2.59 0.29 – 11.43 | -2.96 ± 3.55 -20.77 – 15.80 |
| 2018 | Bathurst Inlet | 3215 | 5.80 ± 0.91 2.84 – 7.51 | 21.86 ± 1.91 19.81 – 27.52 | 305.39 ± 5.79 293.75 – 322.70 | 0.37 ± 0.15 -0.01 – 0.84 | 8.89 ± 2.27 4.79 – 14.69 | -17.58 ± 8.01 -42.85 – -5.30 |
| | Dease Strait West | 1516 | 3.28 ± 1.80 -1.29 – 6.03 | 26.83 ± 1.01 24.42 – 28.50 | 298.91 ± 18.93 250.68 – 386.92 | 0.39 ± 0.25 0.06 – 1.30 | 8.38 ± 3.07 1.88 – 13.16 | -17.89 ± 10.91 -44.92 – -0.60 |
| | Wellington Bay | 1414 | 3.03 ± 1.21 1.23 – 6.13 | 26.73 ± 0.80 24.48 – 27.93 | 268.16 ± 8.60 253.76 – 294.05 | 0.20 ± 0.11 -0.16 – 0.46 | 6.85 ± 2.12 0.28 – 11.90 | -17.72 ± 7.68 -42.44 – -7.81 |
| | Finlayson Islands | 1352 | 3.23 ± 1.47 0.47 – 5.82 | 26.62 ± 1.02 24.68 – 28.07 | 284.96 ± 16.21 248.81 – 317.35 | 0.24 ± 0.11 -0.07 – 0.62 | 8.29 ± 2.36 1.41 – 12.32 | -21.20 ± 8.83 -46.82 – -7.29 |

| Year | Location | | | | | | | |
|---|---|---|---|---|---|---|---|---|
| | Cambridge Bay | 972 | 5.02 ± 1.88 1.34 – 8.14 | 23.80 ± 3.11 17.66 – 27.95 | 253.52 ± 20.43 217.83 – 301.45 | 0.21 ± 0.11 -0.27 – 0.62 | 6.23 ± 1.98 2.33 – 11.76 | -15.35 ± 9.89 -51.97 – -2.72 |
| | Queen Maud Gulf | 1043 | 3.53 ± 0.89 1.87 – 5.72 | 27.06 ± 1.17 21.56 – 28.20 | 286.50 ± 15.82 250.61 – 310.12 | 0.18 ± 0.13 -0.19 – 0.45 | 4.70 ± 1.66 1.61 – 10.31 | -7.58 ± 5.98 -34.36 – -0.99 |
| | Average all | 9512 | 4.29 ± 1.79 -1.29 – 8.14 | 24.82 ± 2.83 17.66 – 28.50 | 288.55 ± 22.12 217.83 – 386.92 | 0.29 ± 0.18 -0.27 – 1.30 | 7.70 ± 2.71 0.28 – 14.69 | -16.79 ± 9.34 -51.97 – -0.60 |
| 2019 | Wellington Bay | 718 | 6.78 ± 0.97 3.81 – 8.56 | 19.81 ± 1.79 16.08 – 23.35 | 353.43 ± 14.76 320.60 – 394.70 | 0.22 ± 0.02 0.17 – 0.28 | 1.92 ± 0.70 0.64 – 2.64 | -0.62 ± 0.25 -1.03 – -0.01 |
| | Finlayson Islands | 2870 | 7.37 ± 0.96 4.74 – 9.65 | 21.72 ± 1.24 18.13 – 24.21 | 392.03 ± 18.97 320.90 – 427.40 | 0.20 ± 0.04 0.11 – 0.31 | 4.82 ± 2.35 0.64 – 7.47 | 0.02 ± 0.49 -1.89 – 2.35 |
| | Cambridge Bay | 1097 | 7.20 ± 0.55 5.21 – 8.81 | 20.03 ± 2.05 12.23 – 22.61 | 359.27 ± 17.62 308.66 – 418.43 | 0.21 ± 0.04 0.08 – 0.32 | 2.63 ± 1.46 0.71 – 4.50 | -0.95 ± 0.89 -3.20 – 0.21 |
| | Queen Maud Gulf | 6192 | 6.72 ± 1.29 2.86 – 8.81 | 19.47 ± 1.79 13.07 – 24.54 | 369.83 ± 11.04 327.32 – 404.64 | 0.26 ± 0.07 0.11 – 0.52 | 2.60 ± 1.44 0.10 – 5.87 | -0.75 ± 0.62 -3.41 – 0.10 |
| | Average all | 11058 | 6.96 ± 1.16 2.86 – 9.65 | 20.12 ± 1.93 12.23 – 24.54 | 373.37 ± 18.75 308.66 – 427.40 | 0.24 ± 0.07 0.08 – 0.52 | 3.13 ± 1.96 0.10 – 7.47 | -0.57 ± 0.69 -3.41 – 2.35 |

## 4. Discussion

Presented in the results above are the multiyear summertime $p\mathrm{CO}_{2\,(sw)}$ observations made on *RV Martin Bergmann*. These
data reveal the spatial and inter-annual variability of $p\mathrm{CO}_{2\,(sw)}$ throughout the open-water season in the Kitikmeot Sea. To
maximise the value of the $p\mathrm{CO}_{2\,(sw)}$ observations made on *RV Martin Bergmann* we will now present and discuss these new
measurements alongside previous measurements and in the context of our current understanding of the carbonate system in
the region.

**4.1 Local scale – comparisons with the ocean carbon observatories**

The two local observatories, the ONC mooring in Cambridge Bay and the Qikirtaarjuk Island observatory (Figure 1), provide measurements throughout the year that are not possible with shipboard observations. $p\mathrm{CO}_{2\,(sw)}$ is directly measured on the ONC mooring, whereas $p\mathrm{CO}_{2\,(sw)}$ is calculated from the flux derived using measurements from the Qikirtaarjuk Island observatory eddy covariance "EC tower". Using the $p\mathrm{CO}_{2\,(sw)}$ observations from these two observatories alongside the new *RV Martin Bergmann* measurements allows us to construct a multiyear timeline of $p\mathrm{CO}_{2\,(sw)}$ in the region (Figure 5). It

should be noted that the three measurement sources in Figure 5 are not co-located, the Qikirtaarjuk Island observatory on the Finlayson Islands is 35 km west of the ONC mooring (Figure 1) and the Bergmann measurements span a slightly wider area (Figure 2). Despite the spatial disparity in these measurements, it should be acknowledged that for calculations of global $\mathrm{CO}_2$ flux on a 1° x 1° grid, the majority of these measurements would fall within the same grid cell. It might be expected that on these sorts of spatial scales the measurements should agree closely, but that is not always the case (Figure 5).


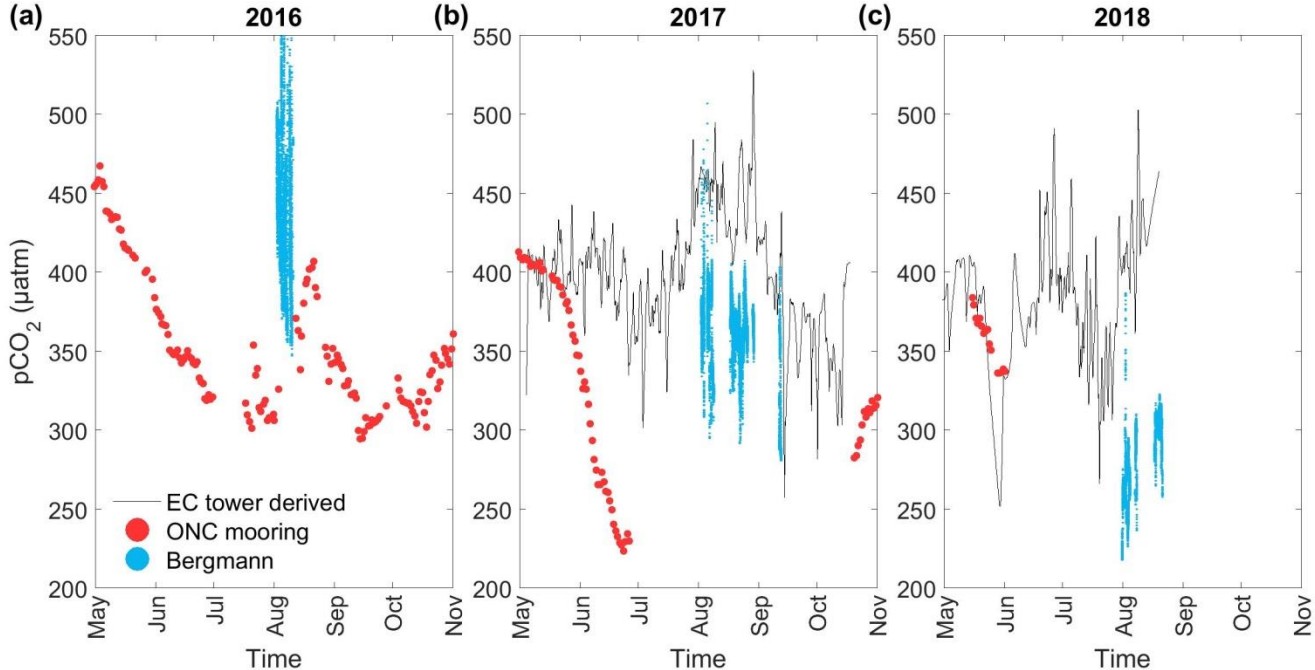

**Figure 5: Surface $p\mathrm{CO}_{2\,(sw)}$ from across the Kitikmeot Sea made in (a) 2016, (b) 2017 and (c) 2018. $p\mathrm{CO}_{2\,(sw)}$ measurements from the ONC mooring are shown as red dots, all $p\mathrm{CO}_{2\,(sw)}$ measurements from the *RV Martin Bergmann* are shown as blue dots and $p\mathrm{CO}_{2\,(sw)}$ inferred from Eddy covariance at the Qikirtaarjuk Island observatory are shown as a black line.**

The *RV Martin Bergmann* $p\mathrm{CO}_{2\,(sw)}$ data are lower in 2017 (Figure 5b) and 2018 (Figure 5c) relative to the values predicted from the EC tower, even when measurements were made in the footprint of the EC tower. For example, from 18:30 – 23:10 on 3 August 2017, $p\mathrm{CO}_{2\,(sw)}$ from the EC tower was 415 µatm and from *RV Martin Bergmann* was 390 µatm; whereas, from

05:50– 06:40 on 1 August 2018, $pCO_{2 (sw)}$ from the EC tower was 409 µatm and from *RV Martin Bergmann* was 262 µatm. Accounting for a thermal skin temperature of 0.17°C in the *RV Martin Bergmann* data only alters the $pCO_{2 (sw)}$ by about ~ 3 µatm based on the 4.23% °C$^{-1}$ Takahashi et al. (1993) constant. For the *RV Martin Bergmann* $pCO_{2 (sw)}$ to match values from the EC tower, based on the the 4.23% °C$^{-1}$ constant the SST at the surface would need to be 1.46 °C greater at the surface on 3 August 2017 and 10.52°C greater at the surface on 1 August 2018 than measured by *RV Martin Bergmann* at 1 m.

Modelling results do not support the existence of temperature differences of the magniture that can account for the $pCO_{2 (sw)}$ difference on 3 August 2017 (Xu et al., 2021). It is possible that the SST measured from the 13 m mooring which is used to calculate pCO$_2$ is not representative of the surface interface, which would bias the schmidt number and $k_0$ used in the calculation of $pCO_{2 (sw)}$ from the EC tower; yet, even if this were the case, the magnitude of the impact can not explain the larger $pCO_{2 (sw)}$ differences between the methods (146 µatm). Even though the *RV Martin Bergmann* measurements are being

made close to the surface (at a depth of 1 m), the most likely explanation for the differences in $pCO_{2 (sw)}$ between the two methods is surface stratification in this upper meter. The impact of surface stratification on $pCO_{2 (sw)}$ has been observed elsewhere in the Arctic (Ahmed et al., 2020;Dong et al., 2021b) including for cases where differences can be up to 200 µatm (Miller et al., 2018). Surface stratification in the Kitikmeot Sea is caused by melting of first-year sea ice and the large freshwater input by rivers (rivers alone can contribute an estimated 70 cm of freshwater to the surface annually; (Williams et

al., 2018). The fact that the EC tower $pCO_{2 (sw)}$ was higher than the *RV Martin Bergmann* $pCO_{2 (sw)}$ would suggest that this is due to river induced stratification, as Arctic riverine water is typically higher in $pCO_{2 (sw)}$ (Cai et al., 2010), indeed this was true between the 30 June and 2 July 2017 for Freshwater Creek (Manning et al., 2020). Interestingly, the predicted $pCO_{2 (sw)}$ from the EC tower shows a peak in early August 2017 and a downwards trend through to the end of August, something that is also seen in the ship-based $pCO_{2 (sw)}$ observations (Figure 5b). Similarly, the predicted $pCO_{2 (sw)}$ from the EC tower

increases in August 2018 at a similar rate to the increase seen in the shipboard $pCO_{2 (sw)}$ observations (2.22 µatm d$^{-1}$;Figure 5c). The fact that similar trends can be observed in the *RV Martin Bergmann* and the EC tower $pCO_{2 (sw)}$ does suggest that seasonal trends in the region are detectable with both methods. However, the general disagreement between the *RV Martin Bergmann* measurements and those from the EC tower highlights the need for year-round $pCO_{2 (sw)}$ observations in the flux footprint of the EC tower. Additionally, interfacial $pCO_{2 (sw)}$ measurements and vertical profiles may help reconcile the

observed disparities seen between the two measurement sources of data.

There is good agreement in the $pCO_{2 (sw)}$ values between the EC tower and the ONC mooring in May, June, and October 2017 (Figure 5b) and in May and June 2018 (Figure 5c). The breakdown of stratification at the end of the ice-free summer period and over the winter (Xu et al., 2021) may explain the good agreement between the EC tower and the ONC mooring at

these times. In June 2017, the two systems diverge. Specifically, the $pCO_{2 (sw)}$ at the ONC mooring decreases due to a spring bloom (Duke et al., 2021), whereas $pCO_{2 (sw)}$ from the EC tower is not impacted, as the bloom in Cambridge Bay is caused by wastewater discharge (Back et al., 2021) it might be expected that this signal would not detectable at the EC tower.

There appears to be some agreement between the *RV Martin Bergmann* collected data and the ONC mooring in the summer of 2016. Unfortunately, the servicing period of the ONC mooring overlapped with the *RV Martin Bergmann* cruise dates meaning there was no period of direct overlap between the two data sets. The four periods when the *RV Martin Bergmann* was moored up within 0.5 km of the mooring on 5 August 2016 05:20 to 5 August 2016 11:10, 7 August 2016 05:40 to 8 August 2016 01:20, 9 August 2016 08:20 to 9 August 2016 14:30, 10 August 2016 00:50 to 10 August 2016 21:40 the average $pCO_{2 (sw)}$ values were 433, 421, 406 and 406 µatm respectively. $pCO_{2 (sw)}$ at the ONC mooring on 3 August 2016 10:00 was 326 µatm and on 12 August 2016 12:40 was 371 µatm. Disagreement between the ONC mooring and the *RV Martin Bergmann* here may be due to the different intake depths of the two systems. Stratification may mean the ONC mooring is not always representative of $pCO_{2 (sw)}$ closer to the air−sea interface, especially during parts of ice free period; however, CTD profiles from 2018 do indicate there is stratification in the surface 10 m in the summer (Back et al., 2021). The spring 2016 measurements from the ONC mooring show that $pCO_{2 (sw)}$ was high in the spring leading into the summer field season, and the trend towards increasing $pCO_{2 (sw)}$ due to warming was captured in August 2016 by both the ONC mooring and the *RV Martin Bergmann* observations.

Combining the data sources in this way highlights the value of having these different observatories to look at multiyear changes. The observatories provide context to the variability in the summertime $pCO_{2 (sw)}$ measurements from local ships. The intermittence of the measurements from the ONC mooring and the Qikirtaarjuk Island observatory reflects the challenges in making these novel measurements in an extreme environment. Knowledge about how to operate both observatories and prevent instrument outages means that future measurements will build towards much needed continuous and complementary multiyear datasets.

## 4.2 Regional scales – spatial variability in the underway data

Focusing back on the *RV Martin Bergmann* data, there is clear evidence of spatial regional variability in the underway data. $pCO_{2 (sw)}$ was typically lower by ~20−40 µatm in the small bays (Cambridge Bay, and Wellington Bay) and larger inlets surveyed (Bathurst Inlet, Chantrey Inlet) compared to the central channel (e.g., Dease Strait West, the Finlayson Islands, and Queen Maud Gulf) (Table 1). The reason for relatively lower $pCO_{2 (sw)}$ in the Bays and Inlets is not readily apparent. Using the 4.23 % °C$^{-1}$ constant from Takahashi et al. (1993) it is possible to test whether the pattern of lower $pCO_{2 (sw)}$ in the Bays and Inlets was driven by temperature, for a representative 360 µatm value for $pCO_{2 (sw)}$ to be ~20−40 µatm lower it would need to be between 1.35 and 2.78 °C colder. Rather than being colder, many of these regions, such as Bathurst Inlet, were warmer, and based on the Takahashi et al. (1993) constant, would thus have a predicted higher $pCO_{2 (sw)}$. Although the fluorescence sensor was not calibrated against *in situ* measurements, the fluorescence signal was consistent with previous measurements that showed the region to have widespread low primary production at the surface (Martin et al., 2013). Inspite of the lack of high surface chlorophyll-a concentrations, biological production at depth cannot be ruled out as an explanation for lower $pCO_{2 (sw)}$ in the bays. For example, wastewater discharge has been shown to cause a deep (20 – 30 m) chlorophyll

bloom in Cambridge Bay (Back et al., 2021). A large under ice (Arrigo et al., 2012;Mundy et al., 2009) or ice edge (Perrette et al., 2011) phytoplankton bloom earlier in the season could also explain lower summertime $pCO_{2\,(sw)}$ values in these bays and inlets. It is also possible that these regional differences are driven by regional freshwater inputs; all four identified regions are fed by rivers and there are sharp salinity transitions of ~5 that point to the existence of mixing and fronts (Figure 4c). Rivers are typically thought to be highly supersaturated in $pCO_{2\,(sw)}$ in the Arctic due to organic matter breakdown (Teodoru et al., 2009), potentially contributing to higher $pCO_{2\,(sw)}$ in these bays and inlets. However, whilst local rivers are high in $pCO_{2\,(sw)}$ (Manning et al., 2020), they are typically unbuffered and thus have much lower DIC relative to seawater. Whilst the average values for riverine TA (565 $\mu$mol kg$^{-1}$) and DIC (533 $\mu$mol kg$^{-1}$) in the CAA are low, maximum measured values for TA (2272 $\mu$mol kg-1) and DIC (2252 $\mu$mol kg-1) can be as high or higher than in seawater, depending on the bedrock type underlying the drainage basin (Brown et al., 2020). Dilution by low $pCO_{2\,(sw)}$ ice meltwater does lower $pCO_{2\,(sw)}$ (Cai et al., 2010;Meire et al., 2015), so it may be that sea ice meltwater in these bays and inlets may be contributing to the lower observed $pCO_{2\,(sw)}$.

The ONC mooring is located in Cambridge Bay in shallow water (sensor depth 7 m), at this depth the mooring is not impacted by the Freshwater Creek plume which is detectable at < 2 m (Duke et al., 2021;Manning et al., 2020). It is still unclear how much of an impact being located in the Bay has on the representativeness of these measurements for the Kitikmeot region. As the *RV Martin Bergmann* travelled into and out of the Bay multiple times during the four years of observations, differences in $pCO_{2\,(sw)}$ measured in the Bay and outside the Bay may help identify whether the ONC mooring site is representative of the region as a whole. All transects into and out of Cambridge Bay are shown in Figure 6. Two sub-regions are designated, inside the Bay and outside the Bay, here $pCO_{2\,(sw)}$ from the *RV Martin Bergmann* was averaged every two days for which there were data available (Table 2). As seen in Table 2, $pCO_{2\,(sw)}$ was similar (typically < $\pm$15 $\mu$atm) inside and outside of the bay . On 17 August 2017, $pCO_{2\,(sw)}$ was much higher (39.6 $\mu$atm) in the Bay. As measurements are similar before (8 August 2016 / 9 August 2016) and after (19 August / 20 August), it would appear that this difference is caused by a process only occurring in the Bay; possibly related to the river plume. Overall, the agreement between the measurements inside and outside of the Bay is encouraging and suggests that $pCO_{2\,(sw)}$ in Cambridge Bay, at least broadly agrees with that in the main Channel. Without more information, it is difficult to conclude whether the mooring is truly representative of the wider Kitikmeot Sea.

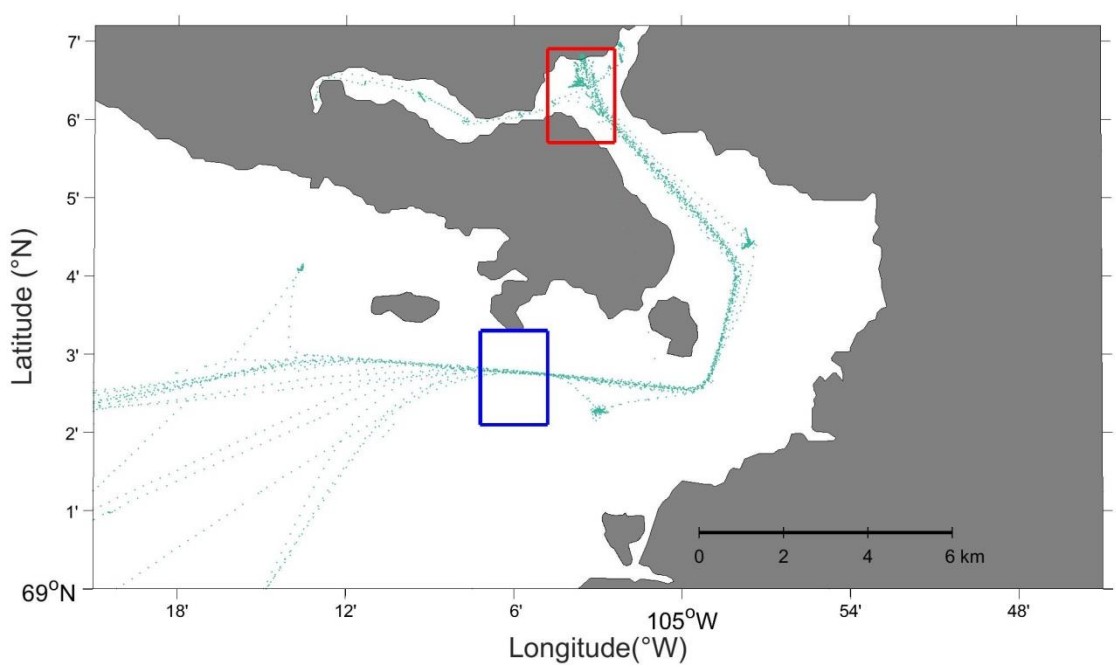

**Figure 6: Zoomed in view showing the location of all the $p\text{CO}_2$ (sw) transects (green) measured in and out of Cambridge Bay during the four years of transects. The regions used to define inside the Bay and outside the Bay are shown by a red and blue box respectively.**

Table 2: Average $p\text{CO}_2$ (sw) measured by the *RV Martin Bergmann* inside and outside of Cambridge Bay.

| Date | $p\text{CO}_2$ (sw) inside Cambridge Bay | $p\text{CO}_2$ (sw) outside Cambridge Bay | $p\text{CO}_2$ (sw) difference (inside Bay –outside Bay) |
|---|---|---|---|
| 5 August 2016 | 482.5 | 450.9 | 31.6 |
| 7 August 2016 to 8 August 2016 | 468.2 | 456.1 | 12.1 |
| 9 August 2016 to 10 August 2016 | 467.4 | 457.7 | 9.7 |
| 4 August 2017 to 5 August 2017 | 375.1 | 371.0 | 4.1 |
| 6 August 2017 to 7 August 2017 | 374.2 | 370.2 | 4.0 |

| | | | |
|---|---|---|---|
| 8 August 2017 to 9 August 2017 | 356.0 | 362.1 | -6.1 |
| 17 August 2017 | 420.9 | 381.3 | 39.6 |
| 19 August 2017 to 20 August 2017 | 371.2 | 374.8 | -3.6 |
| 29 August 2017 | 376.7 | 381.0 | -4.3 |
| 31 July 2018 - 1 August 2018 | 221.5 | 231.3 | -9.8 |
| 2 August 2018 to 3 August 2018 | 251.9 | 246.1 | 5.8 |
| 8 August 2018 | 219.2 | 221.5 | -2.3 |
| 9 August 2019 | 311.8 | 326.7 | -14.9 |
| 18 August 2019 to 19 August 2019 | 360.3 | 364.0 | -3.7 |
| 21 August 2019 | 345.1 | 348.2 | -3.1 |

### 4.3 Interannual variability and large scale seasonal trends

We have identified local scale differences between the $pCO_{2\,(sw)}$ values from the *RV Martin Bergmann*, the ONC, and the Qikirtaarjuk Island observatories and regional scales differences between the bays and inlets and the main channel. However, large differences in the *RV Martin Bergmann* $pCO_{2\,(sw)}$ values occured between years. The measurement start date of all four
cruises spanned a very short window of 10 days (2 August 2016, 2 August 2017, 31 July 2018, 9 August 2019). Ahmed et al. (2019) have established the importance of the timing of sea ice breakup on $pCO_{2\,(sw)}$ values in the CAA. During our study, ice breakup began ~2–6 weeks before the start of these cruises (4 July 2016, 22 June 2017, 15 July 2018, 14 July 2019), which we interpret as exerting one of the main controls of the inter–annual variability in the *RV Martin Bergmann* $pCO_{2\,(sw)}$ data.

The very low $pCO_{2\,(sw)}$ values (289 µatm) observed in 2018 (Table 1) could be caused by a combination of low SST $_{(1m)}$ , springtime $CO_2$ depletion by primary production and recent dilution by sea ice melt (Else et al., 2012;Ahmed et al., 2021;Geilfus et al., 2015) or river runoff (Cai et al., 2010), yet we cannot say with certainty which of these processes was most important in producing these low $pCO_{2\,(sw)}$ values. As the ice breakup was late in 2018 (resulting in samples collected
shortly after breakup), it can be assumed that surface ocean $CO_2$ exchange with the atmosphere was limited by the ice cover

until just before these measurements were made, as sea ice is essentially impermeable to gases (Loose et al., 2011;Butterworth and Else, 2018). Additionally, the presence of sea ice through to the end of July in 2018 meant there was far less warming of the surface seawater (average SST $_{(1m)}$ = 4.32 °C), this explaination rules out surface cooling lowering SST $_{(1m)}$ and thus $pCO_{2 (sw)}$. Light penetrating through sea ice between March and June could have driven primary production below and within the ice (Else et al., 2019). Indeed, an increase in under-ice chlorophyll $a$ concentration together with a draw-down of surface nutrients between April to June 2018 indicate under-ice phytoplankton production during this period (Dalman et al., 2019). However, chlorophyll $a$ concentrations did not exceed 0.6 µg L$^{-1}$, as production is limited by surface nutrient availability in the region (Back et al., 2021). It is likely that the melting sea ice stratified the surface and diluted surface $pCO_{2 (sw)}$ as has been observed in other parts of the Arctic (Miller et al., 2018;Ahmed et al., 2020); low surface ocean salinity values in the first weeks of the survey support this. Measurements several weeks into the 2018 cruise show that $pCO_{2 (sw)}$ increased quickly in the following weeks (to ~300 µatm), likely due to a combination of air–sea exchange and the observed surface warming.Interestingly, Ahmed et al. (2019) did not observe $pCO_{2 (sw)}$ values below 300 µatm at any point during the five years of passing through the Kitikmeot Sea. Therefore, 2018 could be an anomalously low year for $pCO_{2 (sw)}$, or the discrepancy could highlight the fact that Ahmed et al. (2019) did not make any measurements immediately after sea ice breakup in this region. Furthermore, the discrepancy could be influenced by the difference in sampling depth of the two $pCO_2$ systems between the *CCGS Amundsen* (7 m) and *RV Martin Bergmann* (1 m). The best way to assess the impact of the sampling depth would be to take simultaneous measurements via the ships intake and at the interface as in Ho and Schanze (2020).

The processes driving the changes in $pCO_{2 (sw)}$ that have been discussed above can be partially quantified using back of the envelope calculations with several assumptions. The individual impact on $pCO_{2 (sw)}$ of dilution by melting sea ice, air–sea gas exchange, net community production (NCP) and warming of seawater are explored across the region for the month of August in 2018.

Firstly, the impact of dilution by sea ice melt can be tested by assuming conservative mixing of TA, DIC, and salinity as in (Meire et al., 2015). For the seawater mixing endmember, surface TA (2034.43 µmol kg$^{-1}$) and DIC (1958.82 µmol kg$^{-1}$), SST (-1.38°C) and salinity (28.64) are taken from seawater bottle data on the 18 June 2018 (Duke et al., 2021) alongside surface silicate (4 µmol L$^{-1}$) and phosphate (0.5 µmol L$^{-1}$) from 2018 (Back et al., 2021). Average values from spring 2019 for TA (356.60 µmol kg-1), DIC (340.24 µmol kg-1) and salinity (4.56) in first year sea ice are used for the sea ice mixing end member (Else et al., 2022). Taking a sea ice thickness of 1.8 m and assuming water expands 10% when it freezes to form sea ice, would suggest melting all the sea ice would add 1.64 m of water, to reach the final salinity of 24.82 (the average recorded value from the *RV Martin Bergmann* measurements) with conservation of salinity would require this freshwater to mix with 8.68 m of seawater. The ratio of these two depths can then be used to provide the predicted TA (1768.26 µmol kg$^{-1}$), and DIC (1702.05 µmol kg$^{-1}$), for the seawater at a salinity of 24.82. Using CO2SYS (Lewis et al.,

1998;Van Heuven et al., 2011) the calculated $pCO_{2\,(sw)}$ value for the initial seawater condition is 369 µatm and after the melting of sea ice $pCO_{2\,(sw)}$ is 302 µatm. The dissociation constants of carbonic acid used in the CO2SYS calculations were those by (Mehrbach, 1973) refit by Dickson and Millero (1987) and the $HSO_4^-$ dissociation constants from (Dickson, 1990). For these calculations temperature was kept constant. As the average measured $pCO_2$ was 289 µatm in 2018, sea ice melt and conservative mixing of seawater can account for the majority (66.75 µatm) of the total change in $pCO_2$ (80 µatm) from the initial seawater conditions in 2018.

Secondly, using the same approach as DeGrandpre et al. (2020) an estimate of the individual and combined impact of air–sea exchange and NCP on $pCO_{2\,(sw)}$ can be made using a simple model with the following assumptions: taking the average flux from the 2018 cruise of -16.79 mmol m$^{-2}$ d$^{-1}$, a 40 m mixed layer depth for Dease Strait (Xu et al., 2021), with a density of ( 996.49 kg m$^{-3}$) from SST (-1.38°C) and salinity (28.64), an upper estimate of NCP (6.63 g C m$^{-2}$) which is the average integrated rate for Cambridge Bay during the open water season of 2018 (Back et al., 2021). With this configuration a change in DIC (+0.0176 µmol kg$^{-1}$ hr$^{-1}$) due to air–sea exchange and NCP (-0.003 µmol kg$^{-1}$ hr$^{-1}$) can be calculated. Taking the combined change in DIC (+0.0142 µmol kg$^{-1}$ hr$^{-1}$) and substituting it into CO2SYS (Van Heuven et al., 2011;Lewis et al., 1998) with the same initial TA, DIC, silicate and phosphate concentrations as on the 18 June 2018, produces a $pCO_{2\,(sw)}$ change of 0.0459 µatm hr$^{-1}$. Scaling this DIC change for the month of August, with no other changes in the system, would increase $pCO_{2\,(sw)}$ by 36.31 µatm (with NCP component reducing $pCO_{2\,(sw)}$ by 9.4 µatm and air–sea exchange component increasing $pCO_{2\,(sw)}$ by 47.34 µatm).

Thirdly, using the 4.23 % °C$^{-1}$ Takahashi et al. (1993) constant, the impact of the 0.078 °C d$^{-1}$ warming trend on $pCO_{2\,(sw)}$ can be calculated for the 22 day period from 31 July 2018 to 22 August 2018. Using the average $pCO_{2\,(sw)}$ value of 289 µatm and SST $_{(1m)}$ of 4.32 °C, an increase in temperature of 1.72 °C would predict a $pCO_{2\,(sw)}$ of 310 µatm. This increase of 21.78 µatm is less than the 22 day increase of 48.84 µatm based on the 2.22 µatm d$^{-1}$ trend in the 2018 *RV Martin Bergmann* data. From this, the impact of warming can account for just under half of the change in $pCO_{2\,(sw)}$, the rest of the increase in $pCO_{2\,(sw)}$ could be due to air–sea gas exchange.

To summarise, modelling the processes impacting $pCO_{2\,(sw)}$ can account for much of the observed changes in $pCO_{2\,(sw)}$ in 2018. Sea ice melt can account for a 66.75 µatm decrease in $pCO_{2\,(sw)}$ equivalent to 83 % of the observed change. The warming of seawater by 1.72 °C in the first 22 days of August would increase $pCO_{2\,(sw)}$ by 21.78 µatm. Air sea gas exchange can account for a 47.34 µatm increase in $pCO_{2\,(sw)}$ in the month of August (34.72 µatm if scaled to the first 22 days). NCP can account for a 9.4 µatm decrease in $pCO_{2\,(sw)}$ in August (-6.7 µatm if scaled to the first 22 days). The actual observed change in $pCO_{2\,(sw)\,in}$ the first 22 days of August was 48.77 µatm which is extremely close to the combined $pCO_{2\,(sw)}$ change from these three processes 48.68 µatm.

While not as heavily undersaturated as in 2018, $pCO_{2\,(sw)}$ was still undersaturated with respect to atmospheric values in both 2017 and 2019. In these two years, measurements were made ~4–8 weeks after sea ice breakup and $pCO_{2\,(sw)}$ values were in the ~350–390 µatm range. Having been ice free for longer, $SST_{(1m)}$ was 3–4 °C warmer in 2017 and 2019 which accounts for much of the $pCO_{2\,(sw)}$ difference relative to 2018. Increased $SST_{(1m)}$ in 2017 and 2019 and a gradual increase in surface salinity in 2019 mirror the seasonal trends seen in Ahmed et al. (2019) where the CAA becomes saltier and warmer over the summer. The 2017 and 2019 $pCO_{2\,(sw)}$ values are similar but still slightly lower than the $pCO_{2\,(sw)}$ values observed in Coronation Gulf by Ahmed et al. (2019) which again likely reflects the slightly earlier sampling period of this study, where undersaturated surface waters that are recently ice-free have not had long to equilibrate with the atmosphere or wam up.

Measured $pCO_{2\,(sw)}$ was much higher in 2016 (445.08 µatm) compared to 2017 and 2019 around four weeks after sea ice breakup. Ahmed et al. (2019) also observed similiar $pCO_2$ supersaturation (464.5 µatm) in the region in 2016 when they made their observations ~2 weeks later. $pCO_{2\,(sw)}$ supersaturation requires either the upwelling of high $pCO_{2\,(sw)}$ deep waters, net heterotrophy, or for $pCO_{2\,(sw)}$ to be close to equilibrium with the atmosphere and then for the seawater to subsequently warm (Chierici et al., 2011). The most plausible of these is the warming of the surface waters. However, if $SST_{(1m)}$ variability was the main factor controlling $pCO_{2\,(sw)}$, it is not apparent why there would be supersaturation in 2016, but not in 2017 and 2019 which were both warmer years. The sea ice breakup time in 2016 was similar to both 2017 and 2019, suggesting that the timing of breakup was also not the only determining factor. We propose that the high $pCO_{2\,(sw)}$ values observed in 2016 may point to the importance of surface $pCO_{2\,(sw)}$ values set in the previous autumn and wintertime modulation of $pCO_{2\,(sw)}$. To determine what processes are altering $pCO_{2\,(sw)}$ between summertime field seasons would require year round sampling or a biogeochemical model, run over multiple years, which are outside of the scope of this paper.

Clearly, many interacting processes are involved in determining surface ocean $pCO_{2\,(sw)}$ values in the Kitikmeot Sea, and as such, predicting surface ocean $pCO_{2\,(sw)}$ in this region is difficult. Ahmed et al. (2019) proposed a model for $pCO2_{(sw)}$ in the CAA as a function of weeks since ice breakup, their model underestimated $pCO_{2\,(sw)}$ in the Kitikmeot Sea by ~26 µatm which they suggest may be due to the influence of rivers. Following their approach, the surface $pCO_{2\,(sw)}$, SST, and salinity measurements from this study are presented as a function of time since ice melt (when sea ice concentration declines below 85%; Figure 7). The *RV Martin Bergmann* observations are broadly consistent with the general $pCO_2$ model of Ahmed et al. (2019), where low $pCO_{2\,(sw)}$ values (~300µatm) are seen shortly after sea melt and higher values (~300-350 µatm) are seen in the two months after sea ice melt. However, the 2016 $pCO_{2\,(sw)}$ values are much higher and the 2018 values are much lower than predicted by the model. The model is also not a good predictor of the observed salinity values in 2016 and 2019. The CAA flux estimate (Ahmed and Else, 2019) determined using the Ahmed et al. (2019) model remains the best estimate for the region. However, the model is clearly unable to capture the full inter–annual variability in the *RV Martin Bergmann* observations. This could be because as a CAA wide model it is not tuned to the Kitikmeot Sea where freshwater inputs are

greater. Fitting a quadratic equation to the *RV Martin Bergmann* $pCO_{2\,(sw)}$ observations produces the following equation: $pCO_{2\,(sw)} = -1.7452(X^2) + 26.0281(X) + 272.7442$ which can be used to model $pCO_{2\,(sw)}$, where X is weeks since ice breakup.

Both models predict very similar $pCO_{2\,(sw)}$ in the first seven weeks after sea ice breakup, the average difference between the models for this period is 8.01 µatm. The models differ more after 7 weeks after sea ice breakup. At 14 weeks after sea ice breakup, the model of Ahmed et al. (2019) predicts a $pCO_{2\,(sw)}$ that is 81.2 µatm higher than the model fit to the *RV Martin Bergmann* $pCO_{2\,(sw)}$ observations. Fundamentally, understanding the drivers of the large interannual variability in $pCO_{2\,(sw)}$ seen in the Kitikmeot Sea requires an understanding of the interconnected processes involved and their timing. The

interannual variability SST $_{(1m)}$ and salinity are comparable to the modelling results of Xu et al. (2021). Expanding on that modelling work with a complex biogeochemical model that can incorporate all the known processes impacting $pCO_{2\,(sw)}$ may make it possible to accurately reproduce the $pCO_{2\,(sw)}$ observations in this region.

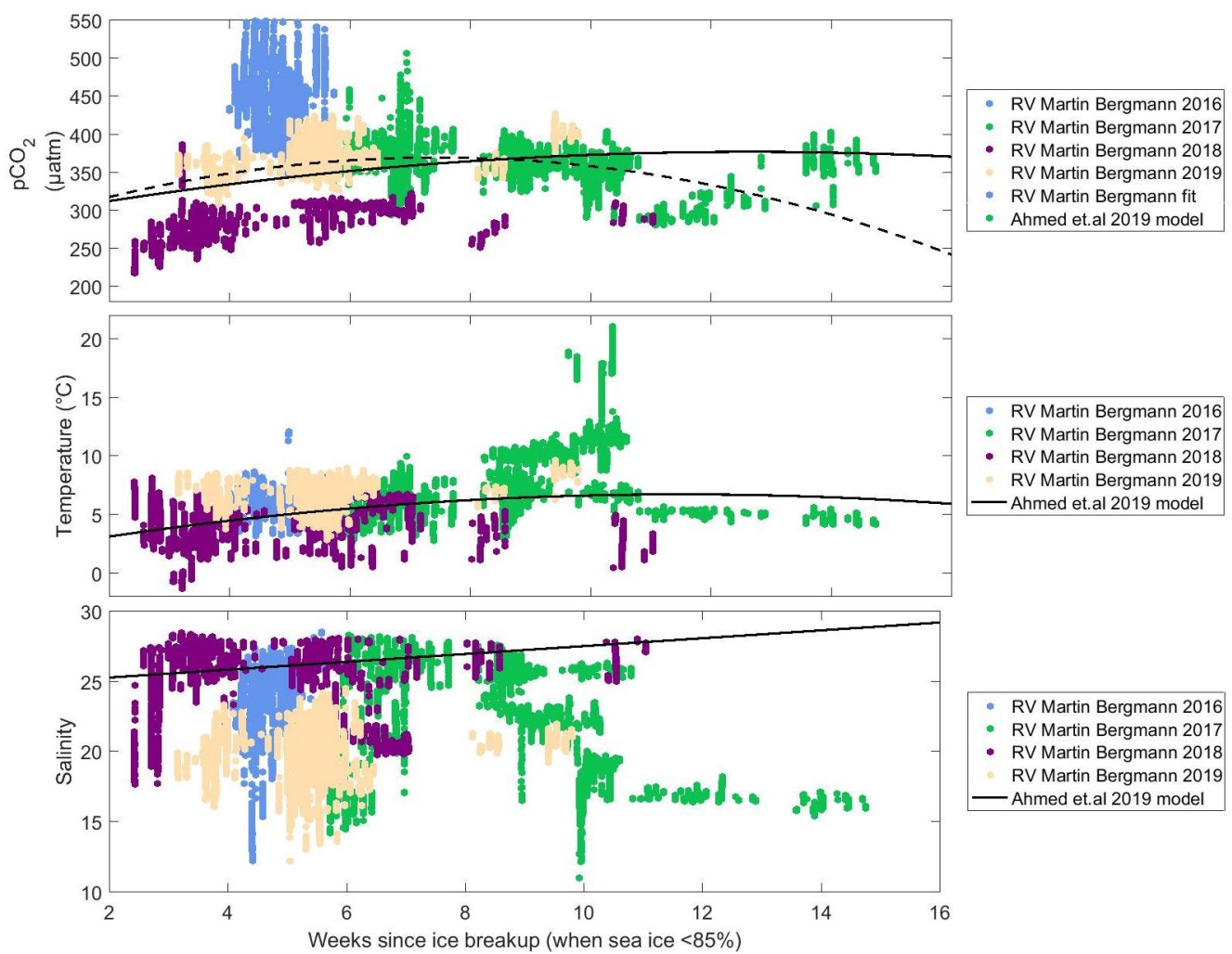

**Figure 7: Surface (a) $p$CO$_{2\ (sw)}$, (b) SST, and (c) salinity from the *RV Martin Bergmann* as a function of weeks of open water for years 2016 to 2019. A quadratic fit to the *RV Martin Bergmann* $p$CO$_{2\ (sw)}$ data is shown as a dashed black line. Black curves represent the model output of (Ahmed et al., 2019)**

## 4.4 The Kitikmeot Sea as a sink for atmospheric CO$_2$

The *RV Martin Bergmann* $p$CO$_{2\ (sw)}$ measurements indicate that the region is a CO$_2$ sink in early August, most years (Table 1). At sea ice breakup, SST $_{(1m)}$ values are low and there are large $\Delta p$CO$_2$ gradients between the surface ocean and the atmosphere, these conditions persist for several weeks after sea ice breakup. Warming of the surface ocean is the likely cause of $p$CO$_{2\ (sw)}$ supersaturation in some years, resulting in the region becoming a net source later in the season. Decreasing SST $_{(1m)}$ at the end of the ice-free season lowers $p$CO$_{2\ (sw)}$ producing a second period when there are larger $\Delta p$CO$_2$ gradients between the ocean and the atmosphere, this is partially identifiable in the *RV Martin Bergmann* measurements from late in 580  2017. The size of the CO$_2$ sink throughout the summer, appears not only to be driven by time since ice breakup, but also by

the absolute surface ocean $pCO_{2\ (sw)}$ value at the time of ice breakup. Ahmed and Else (2019) used remote sensing products to identify this region as a net sink when the flux is integrated over the full ice-free period, our measurements corroborate these findings.

The large variability in $pCO_{2\ (sw)}$ measured in the four years of observations highlights the fact that, in the Arctic, single cruises in only part of the ice-free season are likely not capturing the full seasonal variability. Many $pCO_{2\ (sw)}$ observations in the Arctic are temporally biased towards the middle of the ice-free season, when moving vessels through the Arctic Ocean is easiest. As these single cruises are the only measurements in many of these regions in databases like SOCAT (Bakker, 590    2016), they could result in biased regional flux estimates. In particular, it should be acknowledged that the majority of the CAA is not included in the state of the art observational based products (Landschützer et al., 2020).

## 5. Conclusions

The ONC mooring and EC tower both provide similar $pCO_{2\ (sw)}$ values in spring and autumn showing good agreement between the two platforms. Measured $pCO_{2\ (sw)}$ from the EC tower was sometimes similar to what was measured from the 595    *RV Martin Bergmann* whereas at other times it was much higher. Similar seasonal trends which are likely related to temperature were seen in $pCO_{2\ (sw)}$ from the EC tower and the *RV Martin Bergmann* . Comparing measurements collected by the *RV Martin Bergmann* in and out of Cambridge Bay indicates that Cambridge Bay surface ocean $pCO_{2\ (sw)}$ is similar to that in Dease Strait in August.

The Kitikmeot Sea was a $CO_2$ sink or a very weak $CO_2$ source over the summers of 2016 – 2019, consistent with previous measurements from Ahmed and Else (2019). The $CO_2$ sink was highly variable from year to year at the beginning of August (average observed fluxes of  +3.58, -2.96, -16.79 and -0.57 mmol m$^{-2}$ d$^{-1}$ during the 2016, 2017, 2018, and 2019 cruises respectively) with average $pCO_{2\ (sw)}$ as low as 288.55 µatm and as high as 445.08 µatm. $pCO_{2\ (sw)}$ was much lower in 2018 due to the much lower SST $_{(1m)}$ that year. The magnitude of the air−water $\Delta pCO_2$ throughout the summer appears to be 605    controlled by the absolute $pCO_{2\ (sw)}$ value at the time of ice breakup. Low $pCO_{2\ (sw)}$ values increase in August due to exchange with the atmosphere and warming broadly following the predicted trends using the model developed by Ahmed et al. (2019). In years where $pCO_{2\ (sw)}$ is high when ice breakup occurs, warming can cause a period of slight $pCO_{2\ (sw)}$ supersaturation in summer, in these situations the magnitude of this supersaturation is likely moderated by the air–sea flux reducing $pCO_{2\ (sw)}$. $pCO_{2\ (sw)}$ was found to be ~20−40 µatm lower in the Bays and Inlets that were surveyed; this could be 610    driven by increased freshwater inputs into these isolated regions. Lower $pCO_2$ in the bays and inlets would represent an observational bias in the CAA-wide surveys (Ahmed et al., 2019). Local freshwater fluxes into the southern CAA are much greater than elsewhere in the CAA, meaning that this bias might be more prominent in the Kitikmeot Sea. Further observations in these regions may complement the basin-level $pCO_2$ mapping.

These findings provide a more nuanced picture of the considerable inter-annual variability in $p\text{CO}_{2\,(sw)}$ observed during repeat cruises in the same region, underscoring how much may be missed by relying on data collected during one-off cruises along the dynamic Arctic coasts. In particular, the $p\text{CO}_{2\,(sw)}$ at the time of ice melt is very important as it dictates the magnitude and direction of the flux for much of the ice-free period; however, a better understanding of $p\text{CO}_{2\,(sw)}$ through the ice covered period is needed to help unravel the seasonal and interannual variability.

## 6. Acknowledgements

Parts of this research were completed on or adjacent to Inuit Owned Lands under the authority of the Nunavut Land Claim Agreement, and the work was licensed by the Nunavut Research Institute. We thank the Ekaluktutiak Hunters and Trappers Organization and the community of Cambridge Bay for their hospitality and support of this project. Richard Sims was supported through the University of Calgary's Eyes High Postdoctoral fellowship program. This work was funded by The Marine Environmental Observation, Prediction and Response Network (MEOPAR, project number 1-02-02-004.1), The Kitikmeot Sea Science Study (K3S), Fisheries and Oceans Canada, the Arctic Research Foundation, the Natural Sciences and Engineering Research Council of Canada (Discovery and Northern Research Supplement grants to B. Else, RGPIN-2015-04780), and the Canada Foundation for Innovation (John R. Edwards Leaders Fund grant to B. Else, 34814). Logistical support was provided by the Arctic Research Foundation. Samantha Jones, Stephen Gonski and Patrick Duke were supported by Polar Knowledge Canada through the Northern Scientific Training Program. We thank Polar Knowledge Canada and the Canadian High Arctic Research Station for their support and for providing accommodation and vehicles during field campaigns. Data from Ocean Networks Canada was accessed under their Creative Commons CC-BY 4.0 License. We thank the captain and crew of the *RV Martin Bergmann* for all their assistance. We also thank Shawn Marriott and Francis Emingak for all their assistance in the field, and Sophia Ahmed for her work on data interpretation.

## 7. Author Contributions

This manuscript was written by RPS, all co-authors made contributions to the final paper. BTGE installed the underway system at the start of each field season. The performance of the underway system was monitored by BTGE with help from SFJ, SFG, KAB, PJD and RPS. RPS organised and processed the data. RPS made the figures and interpreted the results and with support from MA and BTGE. BJB analysed the data from the EC tower and provided that data for this paper. PJD provided the data from the ONC mooring. BTGE, KAB, CJM and WJW were central in planning the cruise programme. BTGE oversaw completion of the work.

## 8. Data and code availability

The processed underway data from the *RV Martin Bergmann* which is the new data described in this paper is available in the supplement as .mat files. The raw and processed underway data from the *RV Martin Bergmann* data will also be available via Zenodo. The final processed data will also be submitted to the Surface Ocean Carbon Atlas (SOCAT). The wind data and inferred seawater $p$CO$_2$ data from the EC tower are included in the supplement as .mat files. The ONC mooring data is freely available at https://data.oceannetworks.ca/home. The AMSR2 sea ice data https://seaice.uni-bremen.de/data/amsr2/asi_daygrid_swath/n3125/ the NCEP winds https://psl.noaa.gov/data/gridded/data.ncep.reanalysis2.html and the atmospheric $p$CO$_2$ data from Barrow ftp://aftp.cmdl.noaa.gov/data/greenhouse_gases/co2/in-situ/surface/ which were used in this paper are all freely available from their online repositories, literature citations are provided in the manuscript text. Processing code and the code needed to reproduce the figures was written in Matlab 2016a. The code is provided in the supplement and is also available at https://github.com/Richard-Sims/Sims_2022_Bergmann_pCO2.

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
