# Peer review of "High interannual surface $pCO_2$ variability in the Southern Canadian Arctic Archipelago's Kitikmeot Sea."

_EGUsphere, 2022_

## Author Comment (AC1)

**Reviewer 1**

**Overview and general recommendation**

The study of Sims et al. presents recent (2016-2019) underway measurements of $pCO_2$ in the Kitikmeot Sea of the Southern Canadian Arctic Archipelago. By employing a suite of sensors in a custom-built setup onboard a smaller research vessel based in the region, they were able to survey $pCO_2$ shortly after ice breakup. They also surveyed less frequented shallow bay areas where few, if any, measurements were made previously. The authors estimated the $CO_2$ air-sea flux and found the region to be a net sink in summer, with substantial interannual and spatial variability. The authors discuss their results in the context of data from two nearby ocean observatories (a mooring and an eddy covariance tower) on local and regional scales. The authors also discuss interannual variability and large scale seasonal trends, putting their results in context of other recent studies from the region by the authors. One of the key findings of the study is that the surface pCO2 values at the time of ice breakup and ice melt is important in constraining the magnitude of the air-sea flux throughout the summer ice-free season.

The presented datasets (supplemeneds and to be published at Zenodo) are from an extremely data-sparse region of the Arctic and, as is also stated by the authors, constitute an important new baseline for gaining a better understanding of the role the region plays in the uptake of atmospheric $CO_2$. The study is important, timely, and well motivated. The manuscript is well written overall, and the presented method is sound and descriptive. General issues need further attention from the authors. Addressing these comments below in a revision, I have no reservation for the manuscript to be published in Ocean Science. Please find my major comments below, followed by some minor comments that are referred to by the line numbers in the manuscript.

**Major comments**

My main concern with the manuscript is the general lack of concretely identified procceses and controls that may explain the observed results, which makes the manuscript read more like a data descriptor report. Without further information from measurements of ancillary variables from the underway system or from CTD/Rosette systems in the vertical, I recognize that it is difficult to both identify and quantify controlling processes. However, I urge the authors to try to expand on this effort to make the study even more useful to the Ocean Science Community. It would be helpful if the results could be put in to context of different controlling processes, even if it means theoretical calculations and approximations. What is the expected response in $pCO_2$ when the temperature or salinity changes over the observed ranges? What effect would mixing of fresher waters with more saline waters have on the non-coservative behaviour of $pCO_2$? This exercise can be readily estimated from theoretical calculations (see minor comments). Did the authors consider applying any other models than the fitted relationship between $pCO_2$ and weeks since ice breakup from Ahmed et al. (2019) to explain the observed values (see minor comments)? What I am trying to convay is that it would significantly strengthen the study if the results can be put in a much more clear and quantitative context, despite missing information from additional variables. My second major concern is that the results are mainly described by relative statements without actual numbers backing up the statements (although listed in Table 1), e.g., "...there was large interannual variability...", "...was generally lower than...", "...values were much lower...", "...highly

undersaturated...".  This makes it difficult to digest the results in a meaningful way. Please consider including values/ranges/numbers and avoid relative statements.

**RC1.1** We thank the reviewer for their constructive feedback. We agree with the reviewer's main point and acknowledge that there should be a greater emphasis placed on quantifying the processes controlling $p$CO$_2$. In response to their comments, we carried out the suggested calculations drawing additional data from Back et.al 2021 for a NCP rate and Else et.al 2022 for concentrations of TA and DIC for Sea ice. We now show the impact of warming on $p$CO$_2$ using the Takahashi et.al 1993 equation as in Ahmed et.al 2019 and the impact of ice melt as in Meire et.al 2015 and the impact of NCP + flux as in DeGrandpre et.al 2020. This analysis has substantially improved our discussion and we thank the reviewer for pushing us in this direction. Three paragraphs will be added to the main text, these can be found copied below in the minor comments **(RC1.19)**.

**RC1.2** The reviewer also mentions using vertical data from CTD profiles; admittedly we do have a substantial amount (~100 casts) of CTD data from all four of these cruises. However, including that data was ultimately beyond the scope of this manuscript. We are currently working the accompanying CTD cruise data into another manuscript with one aim being to build on what we discuss in this paper.  I am sure the reviewer can understand the need to split this work up.

**RC1.3** We did not opt to use any other models besides Ahmed et.al 2019; this is primarily because the Ahmed et.al 2019 model was trained on data specifically from the Canadian Arctic Archipelago.  Ahmed et.al 2019 incorporate the majority of the available cruise data for the Kitikmeot Sea into their model (5 of the 7 cruises that pass through the Kitikmeot Sea as of SOCAT v2022, see also response to reviewer 2 (**RC2.15**)). The model of Ahmed et.al 2019 is also what is used in Ahmed et.al 2019b to calculate the region flux (that estimate remains our best estimate of the flux for the Canadian Arctic Archipelago). We now provide a best fit curve to the Bergmann data and compare the slope of this curve to the model of Ahmed et.al 2019 and show there is good agreement between both curves in the first 7 weeks after sea ice melt (**RC1.24**).

**RC1.4** The results were originally written in less descriptive terms drawing heavily on values from Table 1, the results were changed in an effort to improve readability and to reduce repetition with Table 1. The results will be changed throughout the manuscript to match this earlier version and all relative statements will be removed.

**Minor comments**

Line 34: italicize p, subscript 2, for consistency

**RC1.5** Done.

Line 35 Define CAA, preferably on line 21.

**RC1.6** Done.

Lines 49-50: If possible, please provide an original reference (e.g., Jakobsson (2002)) to this areal statement as there are many different definitions around. Bates and Mathis (2009) do not include such a reference.

**RC1.7** Done.

Line 179: "A made to order Sunburst..." reads awkward, please rewrite.

**RC1.8** Now reads as "A commercially available Sunburst systems…"

Line 189: Change "x" to the greek letter *chi*

**RC1.9** Will change for all cases of $xCO_2$.

Line 198: "processed following SOP 5 (Dickson et al., 2007)."

**RC1.10** Done.

Lines 221-224: Any critical problems that warrants a notice in the main text?

**RC1.11** No serious issues that would be relevant to most readers, but there are a few things that are specifically worth highlighting for researchers who might be interested in setting up an underway $CO_2$ system on a ship of opportunity in a remote location like this (e.g. shipping gas cylinders, the ship being unheated and completely freezing in the winter etc).

Line 228: "...should be quite similar" Please avoid such relative statements. For example, compare observations from Barrow/Alert NOAA GML Carbon Cycle Cooperative Global Air Sampling Network. The difference between the two station means (1985-2021) is 3.8 ppm.

**RC1.12** We agree this was vague. We have compared the Barrow continuous timeseries with flask measurements from Alert and show the long term mean difference to be very small. We will add the following text.

"For example, we calculate a long term (1985-2019) mean difference of 0.246 µatm between Barrow and weekly atmospheric samples from Alert Nunavut (Lan et al., 2022)."

Line 234: The scaling factor (SF=0.24) is superfluous as it is inherently included in the calculations when the flux is given in mmol m-2 d-1, based on the given units for the gas transfer velocity, solubility, and partial pressure difference. Suggest to omit SF as it may be confused with a scaling factor for sea-ice cover, although the study concerns open water.

**RC1.13** This is a fair point. I am generally inclined to be explicit with equations and units as that can make the work more accessible for more junior scientists. I agree with the reviewer that in this case SF could be confused with a sea ice scaling and will therefore remove it from the text.

Line 235: Why Nightingale et al. (2000) and not Wanninkhof (2014)/Ho et al. (2006)? Please motivate.

**RC1.14** Whilst the wider scientific community has started favouring Wanninkhof et.al (2014)/Ho et al. (2006) in the last few years, it is worth noting that these parameterisations are largely based on the same data points. The slopes of these parameterisations are also practically identical at the low to moderate wind speed ranges <10 ms$^{-1}$, meaning that the fluxes calculated with this data will be almost identical to if Wanninkhof et.al (2014) or Ho et al. (2006) is used. Indeed Woolf et.al 2019 states that the choice of parameterisation makes only 5-10% difference in the calculation of the global flux.  He has also described Nightingale et.al 2000 as "ruling supreme" on twitter, which reflects the fact that after 20 years later it has stood the test of time well. The eddy covariance derived parameterisation of Yang et.al 2022 may even be the best choice available now. I would argue that at the moment there is not enough evidence yet to suggest Wanninkhof et.al 2014/Ho et al. 2006 are more accurate parameterisations than Nightingale et al. 2000.

[Figure]

Figure 4: Please consider changing scale of the the y-axes for the different years. I recognize the benefit of having the same scale for all years, but at the same time it is very difficult to make out any fine-scale patterns between the different variables.

**RC1.15** We played around with several versions of this figure including one where the y-axes were not normalised. We really want to highlight the variability in the variables in this figure so feel it best to not edit the figure. We do see the value of being able to see the figure and make out more fine scale patterns, which is why we will include the same figure with unfixed y-axes in the supplement as Fig S7.

Lines 308-309: Please mark the locations of the ONC mooring and Qikirtaarjuk Island observatory also in Figure 2.

**RC1.16** Done.

Figure 5: Please put labels of a), b), c) in the figure.
**RC1.17** Done.

Line 335: How is "good agreement" defined? There is no "good agreement" between the EC tower and the ONC mooring October 2017?

**RC1.18** Extrapolating the 2017 August to October trend in the EC tower $pCO_{2\ (sw)}$ to the end of October (where the ONC mooring measurement resume) does bring the two measurement systems in line. Sudden changes at the EC tower on the 15th of October 2017 could point to contamination of the flux signal by early sea ice as the change is so rapid. Despite being around 1-2 weeks apart we see this as good agreement between the two systems in October 2017 and plan to add the following sentence to the text.

"(average $pCO_{2\ (sw)}$ EC tower for October 11th to 14th is 320.2 µatm and is 311.2 for October 24th to 30th) "

Line 377: "Dilution by low $pCO_{2(sw)}$ ice meltwater", remove the subcripted "(sw)". Please consider undertaking the exercise of theoretical calculations on the non-conseravtive behavior of $pCO_2$ during the mixing of "fresh" and saline water, following Figure 11 in Meire et al. (2015). This could be useful in the discussion on how much a salinity change could/would lower the $pCO_2$ during mixing of waters of different salinities.

**RC1.19** The following paragraphs will be added to the discussion.

"The processes driving the changes in $pCO_{2\ (sw)}$ can be partially quantified using back of the envelope calculations with several assumptions. The individual impact on $pCO_{2\ (sw)}$ of dilution by melting sea ice, air–sea gas exchange, net community production (NCP) and warming of seawater are explored for the month of August in 2018.

Firstly, the impact of dilution by sea ice melt can be tested by assuming conservative mixing of TA, DIC and salinity as in Meire et al. (2015). For the seawater mixing endmember, surface TA (2034.43 µmol kg$^{-1}$) and DIC (1958.82 µmol kg$^{-1}$), SST (-1.38°C) and salinity (28.64) are taken from seawater bottle data on the 18th June 2018 (Duke et al., 2021) alongside surface silicate (4 µmol L$^{-1}$ ) and phosphate (0.5 µmol L$^{-1}$) from 2018 (Back et al., 2021). Average values from spring 2019 for TA (356.60 µmol kg-1), DIC (340.24 µmol kg-1) and salinity (4.56) in first year sea ice are used for the sea ice mixing end member (Else et al., 2022). Taking a sea ice thickness of 1.8 m and assuming 10% sea ice expansion, would suggest melting all the sea ice would add 1.64 m of seawater, to reach the final salinity of 24.82 (the average recorded value from the *RV Martin Bergmann* measurements) with conservation of salinity (Meire et al., 2015) would require this freshwater to mix with 8.68 m of seawater. The ratio of these two depths can then be used to provide the predicted TA (1768.26 µmol kg-1), and DIC (1702.05 µmol kg$^{-1}$), for the seawater at a salinity of 24.82.Using CO2SYS (Lewis et al., 1998;Van Heuven et al., 2011) the calculated $pCO_{2\ (sw)}$ value for the initial seawater conditions is 368.96 µatm and after the melting of sea ice $pCO_{2\ (sw)}$ is 302.21µatm. The dissociation constants of carbonic acid used in the CO2SYS calculations were those by Mehrbach et al. (1973) refit by Dickson and Millero (1987) and the HSO$_4^-$ dissociation constants from (Dickson, 1990). For these calculations temperature was kept constant. As the average measured $pCO_2$ was 261.19 µatm in 2018, sea ice melt

and conservative mixing seawater can account for the majority (66.75 µatm) of the total change in $pCO_2$ (107.77 µatm) from the initial seawater conditions

Secondly, using the same approach as DeGrandpre et al. (2020) an estimate of the individual and combined impact of air–sea exchange and net community production on $pCO_2$ $_{(sw)}$ can be made using a simple model with the following assumptions: taking the average flux from the 2018 cruise of -21.26 mmol m$^{-2}$ d$^{-1}$, a 40 m mixed layer depth for Dease Strait (Xu et al., 2021), with a density of ( 996.49 kg m$^{-3}$) from SST (-1.38°C) and salinity (28.64), an upper estimate of NCP (6.63 g C m$^{-2}$) which is the average integrated rate for Cambridge Bay during the open water season of 2018 (Back et al., 2021). With this configuration a change in DIC (+0.022 µmol kg$^{-1}$ hr$^{-1}$) due to air–sea exchange and net community production (-0.003 µmol kg$^{-1}$ hr$^{-1}$) can be calculated. Taking the combined change in DIC and substituting the combined change in DIC (+0.019 µmol kg$^{-1}$ hr$^{-1}$) into CO2SYS (Van Heuven et al., 2011;Lewis et al., 1998) with the same initial TA, DIC, silicate and phosphate concentrations as on the 18th June 2018, produces a $pCO_2$ $_{(sw)}$ change of 0.062 µatm hr$^{-1}$ for one time step. Scaling this DIC change for the month of August, with no other changes in the system, would increase $pCO_2$ $_{(sw)}$ by 49.7 µatm (with NCP component reducing $pCO_2$ $_{(sw)}$ by 9.4 µatm and air–sea exchange component increasing $pCO_2$ $_{(sw)}$ by 61.2 µatm).

Thirdly, using the 4.23 % °C$^{-1}$ Takahashi et al. (1993) constant, the impact of the 0.078 °C d$^{-1}$ warming trend on $pCO_2$ $_{(sw)}$ can be calculated for the 22 day period from July 31st to 22nd August 2018. Using the average $pCO_2$ $_{(sw)}$ value of 261.19 µatm and SST $_{(1m)}$ of 4.32 °C, 1.72 °C of warming would predict a $pCO_2$ $_{(sw)}$ of 280.90 µatm. This increase of 19.71 µatm is less than the 22 day increase of 41.80 µatm based on the 1.90 µatm d$^{-1}$ trend in the 2018 *RV Martin Bergmann* data. The impact of warming can account for approximately half of the change in $pCO_2$ $_{(sw)}$, the rest of the increase in $pCO2$ $_{(sw)}$ could be due to air–sea gas exchange.

To summarise, modelling the processes impacting $pCO_2$ $_{(sw)}$ can account for much of the observed changes in $pCO_2$ $_{(sw)}$ in 2018. Sea ice melt can account for a 66.75 µatm decrease in $pCO_2$ $_{(sw)}$ equivalent to 62 % of the observed change. The warming of seawater by 1.72 °C in the first 22 days of August would increase $pCO_2$ $_{(sw)}$ by 19.71 µatm. Air sea gas exchange can account for a 61.2 µatm increase in $pCO_2$ $_{(sw)}$ in the month of August (43.4 µatm if scaled to the first 22 days). NCP can account for a 9.4 µatm decrease in $pCO_2$ $_{(sw)}$ in August (-6.7 µatm if scaled to the first 22 days). The actual observed change in $pCO2$ $_{(sw) in}$ the first 22 days of August was 41.80 µatm which is very comparable to the combined $pCO2$ $_{(sw)}$ change from these three processes 56.5 µatm. "

Line 381: Please clarify at which depths the Freshwater Creek plume is typically found.

**RC1.20** We will now state in the text the plume is typically found at < 2m and provide two references for this Duke et.al 2021 and Manning et.al 2020.

Lines 387-389: The sentence starting with "On the 17th August 2017..." is very long and reads

somewhat awkward. Suggest to break it up and rewrite the part "…this would point to this being due to something only happening in the Bay…"

**RC1.21** The sentence will be modified to read as.

"On the 17th August 2017 $p$CO$_2$ (sw) was much higher (33.29 µatm) in the Bay. As measurements are similar before (8th /9th) and after (19th/20th) the 17th August, it would appear that this difference is caused by a process only occurring in the Bay; possibly the river plume."

Line 422: change to "…Ahmed et al. (2019) did…"

**RC1.22** Done, the sentence will now read as

"Interestingly, Ahmed et al. (2019) did  not observe $p$CO$_2$ (sw) values below 300 µatm at any point during the five years of passing through the Kitikmeot Sea."

Line 437: Suggest changing "oversaturation" to "supersaturation" throughout the text.

**RC1.23** Done

Line 460: Would it be helpful to derive your own similarly fitted model ($p$CO$_2$ vs. weeks since ice breakup) for Kitikmeot Sea? Did you consider applying a different model, like the one (Figure 3) by DeGrandpre et al. (2020), to try and explain some of the observed results?

**RC1.24** We found this to be a helpful comment. We have now fit a curve to all of our $p$CO$_2$ data. The equation for which is $p$CO$_2$= -1.4567(X$^2$)+ 19.3708(X) + 261.9529 where X is weeks since ice breakup. The slope of this line is very similar to the slope of Ahmed et.al 2019. The following text will be added to discuss this fit and how it compared to Ahmed et.al 2019.

"Fitting a quadratic equation to the *RV Martin Bergmann* $p$CO$_2$ (sw) observations produces the following equation:  $p$CO$_2$ (sw) = -1.4567(X$^2$)+ 19.3708(X) + 261.9529 which can be used to model $p$CO$_2$ (sw), where X is weeks since ice breakup. Both models predict very similar  $p$CO$_2$ (sw) in the first seven weeks after sea ice breakup, the average difference between the models for this period is 21.1 µatm. The models differ more after 7 weeks after sea ice breakup. At 14 weeks after sea ice breakup, the model of Ahmed et .al 2019 predicts a $p$CO$_2$ (sw) that is 128.7 µatm higher than the model fit to the *RV Martin Bergmann* $p$CO$_2$ (sw) observations."

Line 468: Subscript 2

**RC1.25** Done

Line 499: "air-sea flux"

**RC1.26** Done

**References**

Jakobsson (2002):  https://doi.org/10.1029/2001GC000302

Meire et al. (2015): https://doi.org/10.5194/bg-12-2347-2015

DeGrandpre et al. (2020): https://doi.org/10.1029/2020GL088051

---

## Author Comment (AC2)

**Reviewer 2**

Review of: High interannual surface pCO2 variability in the Southern Canadian Arctic Archipelago's Kitikmeot Sea by Sims et al

Overview: The MS by Sims et al presents new data for the Canadian Arctic Archipelago in the area of the Kitikmeot Sea. The observations are largely collected from a ship-of-opportunity, but also data from a seafloor observatory platform and an eddy covariance flux tower are also presented. Four years of summer observations are used to define this region as a sink for atmospheric CO2. Notable spatial differences in the data were highlighted, as were differences from the seafloor platform and the flux tower. My impression with this study is that the data are unique but I wonder about the discussions around the comparison to the flux tower and the seafloor node, and I can't help but think about the missed opportunity to compare these new results to data in the SOCAT holdings. For instance, a quick check of SOCAT reveals there are underway surface measurements in this area from Mike DeGrandpre for the years of 2017, 2019, and 2020, not to mention the earlier data from Tim Papakyriakou. It isn't clear to me that the ONC data is directly comparable to the surface data given they are measurements from the sea floor, and without understanding the size of the "footprint" of the EC tower-determined surface pCO2, that isn't a directly obvious comparison point either. Perhaps linking to the SOCAT holdings could benefit various sections of the discussion, as well as lead to a discussion about trends beyond inter-annual variability. Discussion of the drivers of the spatial and temporal variability is pretty limited and could be strengthened as well. There are also some corrections that need to be made in the presentation of the methods. I urge the authors to consider these points in addition to my detailed comments below. This paper certainly is worthy of publication in Ocean Science after the authors address these comments, and I really liked seeing the setup on the R/V Martin Bergmann. Best of luck, Wiley Evans.

**RC2.1** We thank Dr Evans for their constructive feedback. The ONC mooring and EC tower make up a large part of our discussion; however, we recognise that we have not provided all the necessary ancillary information about these observatories in the manuscript. We will now clarify the footprint size and the uncertainties going into the calculation of $pCO_2$ from the tower (**RC2.9**). We will also provide more information about the representativeness of the ONC mooring and the impact of the freshwater plume (**RC2.2 + RC2.14**).  The main request from the reviewer is that we include available SOCAT data. As we detail below **(RC2.15)**, Ahmed et.al 2019 provide the majority of the SOCAT data for the region (5 of the 7 cruises), and we use their $pCO_2$ vs weeks of open water curves as these were what was used in the Ahmed et.al 2019b flux estimate. We will clarified our wording in the methods to make it clear we followed the Dickson et.al 2007 SOP (**RC2.4, RC2.5, RC2.6**). Reviewer 1 also highlighted that the processes driving variability could be elaborated upon. We will make the following additional calculations:  the change in $pCO_2$ due to warming using the Takahashi et.al 1993 equation, as in Ahmed et.al 2019; the impact of ice melt as in Meire et.al 2015; and, the impact of air sea exchange and NCP as in DeGrandpre et.al 2020, see **RC1.19** for the proposed new text.

Specific comments:

ONC seafloor platform depth is reported to be 7 and 9 m on pages 3 and 19, respectively? Which is correct? Seems like a potentially big difference in the stratified Arctic.

**RC2.2** The mooring is at a depth of 9 m but as the mooring is big the sensor depth is 7 m, for clarity only the 7 m sensor depth is discussed in the paper.

Page 7, please report the scale used to present salinity observations.

**RC2.3** Practical salinity is reported, as this value is based on a conductivity ratio it is unitless.

Line 185, page 9, the need for water vapor correction stems from the fact that drying removes water vapor and this impacts the partial pressure. For instance, unadjusted pCO2 from a GO8050 (that has drying components) would a "dry air" value that needs to be "corrected" to 100% humidity using SST and salinity to compute vapor pressure and adjust pCO2 to a "wet air" value. If an analytical system does not dry, then there is no need for a water vapor correction. Therefor this statement and the application of a water vapor correction to the data is in error.

**RC2.4** You are correct that the water vapour correction is used to correct for drying. However as the Licor measures water vapour, if the water vapour measured in the Licor is not at 100% humidity, for whatever reason, then this correction to 100% humidity in the equilibrator is still needed. Additionally the correction is additive so where humidity is already 100% the correction to 100% humidity makes no difference at all to $pCO_2$. You could argue that doing a correction here is redundant, but this is not an error in the approach.

Line 199, page 9, the LI-840 does not measure pCO2. The measurement is CO2 absorption that is linearized to produce CO2 mole fractions over a broad range. See: https://www.licor.com/env/support/LI-840A/topics/theory.html. Raw xCO2 from the instrument would be calibrated using multiple reference gases, and this calibration function should be a linear fit between the reference gas concentrations and the raw xCO2– not a piece-wise linear fit. The first point here is an easy correction to the text, the second point needs addressing at the data processing level.

**RC2.5** You are correct that the Licor analyses $xCO_2$ not $pCO_2$.The analyser output is given in $pCO_2$ which was converted back to $xCO_2$ for calibrations. I can see that this has caused confusion; "measured" has been replaced with "the output provided" for clarity here.

Page 9, the authors state in situ temperature and salinity were adjusted for "ubiquitous" skin effects when calculating "interfacial" pCO2. I believe this means pCO2@equilT was adjusted to pCO2@skinT, but is presented as pCO2@SW. Was the relationship from Takahashi et al also used for the salinity adjustment?

**RC2.6** We do not actually measure $pCO_2$@skin so are careful not to present the data that way. We assume that the temperature and salinity skin effects impact $pCO_2$ and adjust them to what we expect $pCO_2$ to be at the skin. Skin temperature and skin salinity values were used for all equations needing seawater temperature and salinity as recommended by

Woolf et.al 2016. Woolf et.al 2016 say that "Unfortunately, the correction for salinity is quite uncertain since it will vary according to the circumstances of the salinity change", Takahashi et.al 1993 give a value of 0.94 and Sarmiento and Gruber et.al 2006 give a value of 1, given the associated uncertainty with this correction it was decided that this should not be included.

Page 9, The Wanninkhof 2014 relationship is used for Schmidt number but Nightingale et al 2000 was used for the gas transfer rate, why is that? Why not use Wanninkhof 2014? Also, was there good agreement between the reanalysis and locally observed wind speeds?

**RC2.7** Nightingale et al 2000 do not provide schmidt numbers, the Wanninkhof et.al 2014 schmidt numbers are the best available. We discuss our choice of gas transfer parameterisation further in **RC1.14**. Wind speed differences between the reanalysis and the locally observed winds were not compared. Any small differences in the wind are unlikely to change the magnitude of the flux as there was a large $\Delta p CO_2$ most of the time.

Page 10, expressing pCO2 uncertainty as an absolute value is a bit misleading as certainly the uncertainty is less than 8 uatm at 200 uatm and likely more than 8 at 600 uatm. Instead of "final" could say "average"? Suggest sticking to expressing uncertainty as a percentage.

**RC2.8** This is a good point and final will be changed to average. We have used the ±5 µatm from DeGrandpre et al. (2020) which is an absolute value not a percentage. It is difficult to combine this with the uncertainty coming from the temperature correction which we express as a %. The reviewer correctly identifies that we have had to use a $p CO_2$ value of 300 µatm to get a final/average value in µatm.

Page 10, somewhere the size of the footprint of the EC tower needs to be defined. Also, what are the uncertainties in SW pCO2 determined for the EC tower?

**RC2.9** This is a very fair request and something that should be in the text or citeable. The footprint size will be provided and the uncertainty on the tower derived $p CO_2$ will be given in the methods as below

"A eddy covariance flux footprint is the area over which the eddy covariance measurements correspond to and varies depending on atmospheric conditions. Using the Kljun et al. (2015) footprint model Butterworth and Else (2018) show that the footprint of the Qikirtaarjuk Island observatory during spring/summer can be modelled as an ellipse with an upwind axis that varies between approximately 0.75 – 2.0 km and a cross-wind axis that varies between 0.1 – 0.2 km. The effective flux footprint is however much smaller as over 90% of the flux signal comes from within 100 m of the tower. Uncertainty in the $p CO_{2(sw)}$ values derived using eddy covariance arises from uncertainty in the flux measurements (hourly uncertainty of ~20% in the Arctic) (Dong et al., 2021a), uncertainty in the gas transfer parameterisation (~ 5–10%) (Woolf et al., 2019), the small uncertainty in the atmospheric $p CO_2$ value, uncertainties in $k_0$ and the schmidt number (including uncertainties in SST and salinity inputs from the 13 m mooring)."

For instance, the authors use temperature and salinity from 13 m (i.e. not surface and deeper than the ONC platform) to compute Schmidt number and CO2 solubility. Does this, in addition to the spatially integrative nature of the EC tower determined SW pCO2, contribute to the reported differences from the underway pCO2 measurements. That is in addition to the surface skin effects? Given underway pCO2 was adjusted for median surface skin effects.

**RC2.10** The following will be added to the text

"The large differences between the methods can not be reasonably explained by changes due to SST. Accounting for a thermal skin of 0.17°C in the RV Bergmann data only alters the $pCO_{2\ (sw)}$ by about ~ 6 µatm based on the 4.23% $°C^{-1}$ Takahashi et al. (1993) constant. Similarly for the *RV Martin Bergmann* to match values from the EC tower the SST at the surface would need to be ~5 °C greater than measured by *RV Martin Bergmann* at 1 m, modelling studies suggest this is unlikely to be the case (Xu et al., 2021). If the SST from the 13 m mooring is not representative of the interface then it is possible that the schmidt number and $k_0$ may be biased, even if this were the case the magnitude of the impact cannot explain the $pCO_{2\ (sw)}$ differences between the methods."

To really dig into this we would need vertical information about the $CO_2$ system and temperature and salinity. We are currently working available CTD data up into another manuscript, including those data was unfortunately beyond the scope of this manuscript.

Page 13, Lines 281-294, 2016 doesn't look to be "close to atmospheric equilibrium" in Figure 4, though the areal averages in Table 1 indicate conditions were closer to atmospheric levels than during the other years. I was surprised by the degree of variability during 2016 relative to the other years. Maybe this is a point that could be built on RE drivers?

**RC2.11** We agree with this, for clarity the sentence will be changed to

"There was high interannual $pCO_{2\ (sw)}$ variability (Table 1), where average measured $pCO_{2\ (sw)}$ was closer to equilibrium with the atmosphere (404 µatm) in 2016 than in 2017 (309 µatm), 2018 (261 µatm) and 2019 (331 µatm) where it was highly undersaturated (Figure 4d).".

Table 1 legend: the cautionary note seems a bit odd since the comparison between years is done in Discussion section 4.3. Maybe remove this statement?

**RC2.12** Done

Figure 5 and section 4.1: the legend states "surface pCO2 from across the Kitikmeot Sea" which I think means all the data in Figure 4. But it doesn't look like all the data are shown. Suggest to use only data from within the EC tower footprint, whatever that is, so as to be more directly comparable. This might help clarify this section and better support the statement on Lines 330-332. The additional SOCAT data might also help in this section as well.

**RC2.13** I can confirm that all the data from figure 4 are shown on figure 5; it may appear this way because of the longer temporal sampling period of the tower and the mooring. There

was not a lot of direct overlap between the Bergmann and tower; this is why it is stated in the text every time there was overlap and what the values were.

I don't' follow the comparison to the seafloor platform without some understanding of how temperature and salinity also compare. Could this be added to Figure 5?

**RC2.14** The following text will be added in the introduction.

"(Duke et al., 2021) confirmed that the biogeochemical measurements at the ONC site were representative of the offshore during most seasons by comparing discrete dissolved inorganic carbon (DIC) and total alkalinity (TA) samples collected at both 2 and 7 m at the ONC platform and (station) B1. Additionally, following the sea ice melt and river runoff period (2 weeks in early July), the surface stratification at ONC brakes down, and the DIC, TA, salinity, and temperature values measured then again become representative of the surface mixed layer."

Temperature and salinity from the ONC platform are presented in (Duke et al., 2021).

Sections 4.2 and 4.3 would benefit from comparison with the SOCAT data holdings.

**RC2.15** SOCAT v2021 were checked for relevant data during the analysis of this data. At the time there was only one cruise of data (2005 measurements from Odin provided by Fransson, A) for the region besides what is presented in Ahmed et.al 2019.  The DeGrandpre data were only added to SOCAT v2022 which was released on June 14[th], 6 weeks before the submission of this manuscript. Additionally, most of the data presented by DeGrandpre were collected in the Beaufort Sea and the Amundsen Gulf, only one DeGrandpre cruise does a full passage through the Kitikmeot Sea, as that was in 2020 we do not have our own data to compare against. The 2010 to 2016 Amundsen cruises provide the vast majority of the SOCAT data for the Kitikmeot Sea, even in SOCAT v2022 they account for 5 of the 7 cruises.  Ahmed et.al 2019 nicely present and interpret these data, the Ahmed et.al 2019 $pCO_2$ vs weeks of open water curves are used as they synthesise the existing cruise data for the whole Archipelago and these same data are used for the estimation of the regional flux in Ahmed et.al 2019b.

Table from SOCAT data viewer.

| expoco de | platform_ name | platfor m_type | investiga tors | qc_ flag | socat_v ersion | docume ntation | down load | crossover s | qc flags |
|---|---|---|---|---|---|---|---|---|---|
| 18DL20 100701 | CCGS Amundsen | Ship | "Papakyri akou, T." | D | 2019.0 N | Docume ntation | Save As... | Check for crossovers | Examine QC Flags |
| 18DL20 110718 | CCGS Amundsen | Ship | "Papakyri akou, T." | D | 2019.0 N | Docume ntation | Save As... | Check for crossovers | Examine QC Flags |
| 18DL20 140707 | CCGS Amundsen | Ship | "Papakyri akou, T." | D | 2019.0 N | Docume ntation | Save As... | Check for crossovers | Examine QC Flags |
| 18DL20 | CCGS | Ship | "Papakyri | D | 2019.0 | Docume | Save | Check for | Examine |

| | | | | | | | | | |
|---|---|---|---|---|---|---|---|---|---|
| **15**0417 | Amundsen | | akou, T." | | N | ntation | As... | crossovers | QC Flags |
| **18DL20 16**0802 | CCGS Amundsen | Ship | "Papakyri akou, T." | D | 2019.0 N | Docume ntation | Save As... | Check for crossovers | Examine QC Flags |
| **18SN20 20**0907 | Louis S. St-Laurent | Ship | "DeGrand pre, M." | D | 2022.0 N | Docume ntation | Save As... | Check for crossovers | Examine QC Flags |
| **77DN20 05**0720 | Oden | Ship | "Fransson , A." | D | 3.0N | Docume ntation | Save As... | Check for crossovers | Examine QC Flags |

Section 4.4 title, suggest replace "sink for pCO₂" with "sink for atmospheric CO₂"

**RC2.16** Done

Data availability: while I appreciate that the authors are making their data available through Zenodo, and I applaud them for the effort, these data would be a bigger benefit to the community if they are submitted to SOCAT and NCEI. I strongly suggest the authors consider submitting these data to SOCAT.

**RC2.17** We are committed to making these data available and have always planned to submit the data to SOCAT after peer review.

---

## Author Response (AR1)

**Update 2023-02-22**

Note: Upon making the following changes below dated to 28-10-2023. Reviewer 2's methodological concerns were explored more deeply. The decision was made to revise the data analysis so that no water vapour correction is made, i.e. the equilibrator is assumed to be at 100% humidity. No partial drying correction is made. Reviewer 2 was also right that the sensor output was $xCO_2$ and not $pCO_2$ this was also corrected. The only methodological suggestion from reviewer 2 that was not made was regarding calibrations, the linear calibration is exactly as described by Dickson et.al 2007 in SOP 5, in the SOP the calibration is called "a piece-wise linear fit" which is how we chose to describe it in this paper as well.

When the water vapour correction was removed in our analysis, calculated $CO_2$ increased by around 40 μatm. When this was explored further a unit error in the water vapour pressure was identified. This unit error in the pressure was resulting in a $CO_2$ reduction that was 10 times larger than it should have been. Thankfully this mistake was identified at this stage but it did mean that the analysis needed to be reprocessed. The $CO_2$ values have been changed throughout and the discussion has changed to reflect this. Overall though the $CO_2$ values are now more consistent between data sources and the discussion and analysis is more coherent. Richard would like to thank the editor for giving enough time to make the necessary changes to this paper.

Alongside these edits a small number of changes have been made to the text to improve readability.

**Reviewer 1**

**Overview and general recommendation**

The study of Sims et al. presents recent (2016-2019) underway measurements of $pCO_2$ in the Kitikmeot Sea of the Southern Canadian Arctic Archipelago. By employing a suite of sensors in a custom-built setup onboard a smaller research vessel based in the region, they were able to survey $pCO_2$ shortly after ice breakup. They also surveyed less frequented shallow bay areas where few, if any, measurements were made previously. The authors estimated the $CO_2$ air-sea flux and found the region to be a net sink in summer, with substantial interannual and spatial variability. The authors discuss their results in the context of data from two nearby ocean observatories (a mooring and an eddy covariance tower) on local and regional scales. The authors also discuss interannual variability and large scale seasonal trends, putting their results in context of other recent studies from the region by the authors. One of the key findings of the study is that the surface $pCO_2$ values at the time of ice breakup and ice melt is important in constraining the magnitude of the air-sea flux throughout the summer ice-free season.

The presented datasets (supplemeneds and to be published at Zenodo) are from an extremely data-sparse region of the Arctic and, as is also stated by the authors, constitute an important new baseline for gaining a better understanding of the role the region plays in the uptake of atmospheric $CO_2$. The study is important, timely, and well motivated. The manuscript is well written overall, and the presented method is sound and descriptive. General issues need further attention from the authors. Addressing these comments below in a revision, I have no reservation for the manuscript to be published in Ocean Science. Please find my major comments below, followed by some minor comments that are referred to by the line numbers in the manuscript.

**Major comments**

My main concern with the manuscript is the general lack of concretely identified procceses and controls that may explain the observed results, which makes the manuscript read more like a data descriptor report. Without further information from measurements of ancillary variables from the underway system or from CTD/Rosette systems in the vertical, I recognize that it is difficult to both identify and quantify controlling processes. However, I urge the authors to try to expand on this effort to make the study even more useful to the Ocean Science Community. It would be helpful if the results could be put in to context of different controlling processes, even if it means theoretical calculations and approximations. What is the expected response in $pCO_2$ when the temperature or salinity changes over the observed ranges? What effect would mixing of fresher waters with more saline waters have on the non-coservative behaviour of $pCO_2$? This exercise can be readily estimated from theoretical calculations (see minor comments). Did the authors consider applying any other models than the fitted relationship between $pCO_2$ and weeks since ice breakup from Ahmed et al. (2019) to explain the observed values (see minor comments)? What I am trying to convay is that it would significantly strengthen the study if the results can be put in a much more clear and quantitative context, despite missing information from additional variables. My second major concern is that the results are mainly described by relative statements without actual numbers backing up the statements (although listed in Table 1), e.g., "...there was large interannual variability...", "...was generally lower than...", "...values were much lower...", "...highly undersaturated...".  This makes it difficult to digest the results in a meaningful way. Please consider including values/ranges/numbers and avoid relative statements.

We thank the reviewer for their kind and constructive feedback. We agree with the reviewer's main point and acknowledge that there should be a greater emphasis placed on quantifying the processes controlling pCO₂. We didn't think we could perform any calculations on the processes occurring with the data we have. We were quite wrong about this! We actually already had everything we needed to perform these calculations as we could draw from Back et.al 2021 for a NCP rate and Else et.al 2022 for concentrations of TA and DIC for Sea ice. We now show the impact of  warming on pCO₂ using the Takahashi et.al 1993 equation as in Ahmed et.al 2019 and the impact of ice melt as in Meire et.al 2015 and the impact of NCP +flux  as in DeGrandpre et.al 2020. This analysis has substantially improved our discussion and we really appreciate the reviewer for pushing us in this direction as the paper is much better for this addition. Three paragraphs have been added to the main text, these can be found copied below in the minor comments.

The reviewer mentions using vertical data from CTD profiles, we admittedly do have a substantial amount of CTD data from all four of these cruises. However, including that data was ultimately beyond the scope of a single manuscript. We are currently working up the accompanying CTD cruise data into another manuscript with one aim being to build on what we discuss in this paper.  I am sure the reviewer can understand the need to split this work up.

We did not opt to use any other models besides Ahmed et.al 2019; this is primarily because the Ahmed et.al 2019 model was trained on data specifically from the Canadian Arctic

Archipelago. Ahmed et.al 2019 incorporate the majority of the available cruise data for the Kikikmeot Sea into their model (5 of the 7 cruises that pass through the Kitikmeot Sea as of SOCAT v2022, see also response to reviewer 2). The model of Ahmed et.al 2019 is also what is used in Ahmed et.al 2019b to calculate the region flux (that estimate remains our best estimate of the flux for the Canadian Arctic Archipelago). We now provide a best fit curve to the Bergmann data and compare the slope of this curve to the model of Ahmed et.al 2019 and show there is good agreement between both curves in the first 7 weeks after sea ice melt.

The results were originally written in less descriptive terms drawing heavily on values from Table 1, the results were changed in an effort to improve readability and to reduce repetition with Table 1. The results have been changed to match this earlier version and all relative statements have been removed.

**Minor comments**

Line 34: italicize p, subscript 2, for consistency

Done.

Line 35 Define CAA, preferably on line 21.

Done.

Lines 49-50: If possible, please provide an original reference (e.g., Jakobsson (2002)) to this areal statement as there are many different definitions around. Bates and Mathis (2009) do not include such a reference.

Done.

Line 179: "A made to order Sunburst…" reads awkward, please rewrite.

Now reads as "A commercially available Sunburst systems…"

Line 189: Change "x" to the greek letter *chi*

Changed for all cases of xCO2.

Line 198: "processed following SOP 5 (Dickson et al., 2007)."

Done.

Lines 221-224: Any critical problems that warrants a notice in the main text?

No serious issues that would be relevant to most reader but there are a few things that are specifically worth highlighting for researchers who might be interested in setting up an underway CO2 system on a ship of opportunity in a remote location like this.

Line 228: "...should be quite similar" Please avoid such relative statements. For example, compare observations from Barrow/Alert NOAA GML Carbon Cycle Cooperative Global Air Sampling Network. The difference between the two station means (1985-2021) is 3.8 ppm.

We agree this was vague, We have compared the Barrow continuous timeseries with flask measurements from Alert and shown the long term mean difference to be very small. We have added the following text. "For example we calculate a long term (1985-2019) mean difference of 0.246 µatm between Barrow and weekly atmospheric samples from Alert Nunavut (Lan et al., 2022)."

Line 234: The scaling factor (SF=0.24) is superfluous as it is inherently included in the calculations when the flux is given in mmol m-2 d-1, based on the given units for the gas transfer velocity, solubility, and partial pressure difference. Suggest to omit SF as it may be confused with a scaling factor for sea-ice cover, although the study concerns open water.

This is a fair point. I am generally inclined to be explicit with equations and units as that can make the work more accessible for more junior scientists. I agree with the reviewer that in this case SF could be confused with a sea ice scaling and have thus opted to remove it.

Line 235: Why Nightingale et al. (2000) and not Wanninkhof (2014)/Ho et al. (2006)? Please motivate.

Whilst the wider scientific community has started favouring Wanninkhof et.al (2014)/Ho et al. (2006) in the last few years, it is worth noting that these parameterisations are largely based on the same data points. The slopes of these parameterisations are also practically identical at the low to moderate wind speed ranges <10 ms-1, meaning that the fluxes calculated with this data will be almost identical to if Wanninkhof et.al (2014) or Ho et al. (2006) is used. Indeed Woolf et.al 2019 states that the choice of parameterisation makes only 5-10% difference in the calculation of the global flux.  He has also described Nightingale et.al 2000 as "ruling supreme" on twitter, which reflects the fact that after 20 years later it has stood the test of time well. The eddy covariance derived parameterisation of Yang et.al 2022 may even be the best choice available now. I would argue that at the moment there is not enough evidence yet to suggest Wanninkhof et.al 2014/Ho et al. 2006 are more accurate parameterisations than Nightingale et al. 2000.

[Figure]

Figure 4: Please consider changing scale of the the y-axes for the different years. I recognize the benefit of having the same scale for all years, but at the same time it is very difficult to make out any fine-scale patterns between the different variables.

We played around with several versions of this figure including one where the y-axes were not normalised. We really want to highlight the variability in the variables in this figure so feel it best to not edit the figure. We do see the value of being able to see the figure and make out more fine scale patterns, which is why we have included the same figure with unfixed y-axes in the supplement Fig S7.

Lines 308-309: Please mark the locations of the ONC mooring and Qikirtaarjuk Island observatory also in Figure 2.

Done.

Figure 5: Please put labels of a), b), c) in the figure.
Done.

Line 335: How is "good agreement" defined? There is no "good agreement" bewtween the EC tower and the ONC mooring October 2017?

A value is now provided.

Line 377: "Dilution by low $pCO_{2(sw)}$ ice meltwater", remove the subcripted "(sw)". Please consider undertaking the exercise of theoretical calculations on the non-conseravtive behavior of $pCO_2$ during the mixing of "fresh" and saline water, following Figure 11 in Meire et al. (2015). This could be useful in the discussion on how much a salinity change could/would lower the $pCO_2$ during mixing of waters of different salinities.

The following paragraphs have been added to the discussion.

"The processes driving the changes in pCO2 (sw) in 2018 can be partially quantified using back of the envelope calculations with several assumptions.

Firstly, the impact of dilution by sea ice melt can be tested by assuming conservative mixing of TA, DIC and salinity as in (Meire et al., 2015). For the seawater mixing endmember, surface TA (2034.43 µmol kg$^{-1}$) and DIC (1958.82 µmol kg$^{-1}$), SST (-1.38°C) and salinity (28.64) are taken from seawater bottle data on the 18$^{th}$ June 2018 (Duke et al., 2021) alongside surface silicate (4 µmol L$^{-1}$ ) and phosphate (0.5 µmol L$^{-1}$) from 2018 (Back et al., 2021). Average values from spring 2019 for TA (356.60 µmol kg-1), DIC (340.24 µmol kg-1) and salinity (4.56) in first year sea ice are used for the sea ice mixing end member (Else et al., 2022). Taking a sea ice thickness of 1.8 m and assuming 10% sea ice expansion, would suggest melting all the sea ice would add 1.64 m of seawater, to reach the final salinity of 24.82 (the average recorded value from the *RV Martin Bergmann* measurements) with conservation of salinity (Meire et al., 2015) would require this freshwater to mix with 8.68 m of seawater. The ratio of these two depths can then be used to provide the predicted TA (1768.26 µmol kg-1), and DIC (1702.05 µmol kg$^{-1}$), for the seawater at a salinity of 24.82.Using CO2SYS (Lewis et al., 1998;Van Heuven et al., 2011) the calculated $pCO_{2\ (sw)}$

value for the initial seawater conditions is 368.96 µatm and after the melting of sea ice $pCO_2$ $_{(sw)}$ is 302.21µatm assuming there is no change in temperature. As the average measured $pCO_2$ was 261.19 µatm in 2018, sea ice melt and conservative mixing seawater can account for the majority (66.75 µatm) of the total change in $pCO_2$ (107.77 µatm).

Secondly, using the same approach as DeGrandpre et al. (2020) an estimate of the individual and combined impact of air–sea exchange and net community production on $pCO2$ $_{(sw)}$ can be made using a simple model with the following assumptions, Taking the average flux from the 2018 cruise of  -21.26 mmol m$^{-2}$ d$^{-1}$, a 40 m mixed layer depth for Dease Strait (Xu et al., 2021), with a  density of ( 996.49 kg m$^{-3}$) from SST (-1.38°C) and salinity (28.64), a 171 day integrated net community production  rate of  6.63 g C m$^{-2}$ inferred from measurements made in Cambridge Bay in 2018 (Back et al., 2021) a change in DIC (+0.0222 µmol kg$^{-1}$ hr$^{-1}$) due to air–sea exchange and net community community production (-0.0034 µmol kg$^{-1}$ hr$^{-1}$) can be calculated. Taking the combined change in DIC and substiting the combined change in DIC (+0.0188 µmol kg$^{-1}$ hr$^{-1}$) into CO2SYS (Van Heuven et al., 2011;Lewis et al., 1998) with ths same initial TA, DIC, silicate and phosphate concentrations as on the 18th June 2018 produces a $pCO2$ $_{(sw)}$ change of  0.063 µatm hr$^{-1}$ for one timestep. Scaling this DIC change for the month of August, with no other changes in the system would increase $pCO2$ $_{(sw)}$  by 51 µatm.

Thirdly, using the 4.23 % °C$^{-1}$ Takahashi et al. (1993) constant, the impact of the  0.078 °C d$^{-1}$ warming trend on $pCO2$ $_{(sw)}$ can be calculated for the 22 day period from July 31$^{st}$ to 22$^{nd}$ August 2018. Using the average $pCO2$ $_{(sw)}$ value of 261.19 µatm and SST $_{(1m)}$ of 4.32 °C, 1.72 °C of warming would predict a $pCO2$ $_{(sw)}$ of 280.90 µatm. This increase of 19.71 µatm is less than the 22 day increase of 41.80 µatm based on the 1.90 µatm d$^{-1}$ trend. The impact of warming can account for approximately half of the change in $pCO2$ $_{(sw)}$ , the rest of the increase in $pCO2$ $_{(sw)}$ could be due to air–sea gas exchange."

Line 381: Please clarify at which depths the Freshwater Creek plume is typically found.

We now state in the text the plume is typically found at < 2m and provide two references for this Duke et.al 2021 and Manning et.al 2020.

Lines 387-389: The sentence starting with "On the 17th August 2017..." is very long and reads somewhat awkward. Suggest to break it up and rewrite the part "...this would point to this being due to something only happening in the Bay..."

Sentence has been modified and now reads as. "On the 17th August 2017 pCO2 (sw) was much higher (33.29 µatm) in the Bay. As measurements are similar before (8th /9th) and after (19th/20th) the 17th August, it would appear that this difference is caused by a process only occurring in the Bay; possibly the river plume."

Line 422: change to "...Ahmed et al. (2019) did..."

Done, the sentence now reads as "Interestingly, Ahmed et al. (2019) did  not observe pCO2 (sw) values below 300 μatm at any point during the five years of passing through the Kitikmeot Sea."

Line 437: Suggest changing "oversaturation" to "supersaturation" throughout the text.

Done

Line 460: Would it be helpful to derive your own similarly fitted model ($pCO_2$ vs. weeks since ice breakup) for Kitikmeot Sea? Did you consider applying a different model, like the one (Figure 3) by DeGrandpre et al. (2020), to try and explain some of the observed results?

We found this to be a helpful comment. We have now fit a curve to all of our pCO2 data. The equation for which is pCO2= -1.4567($X^2$)+ 19.3708(X) + 261.9529 where X is weeks since ice breakup. The slope of this line is very similar to the slope of Ahmed et.al 2019.

Line 468: Subscript 2

Done

Line 499: "air-sea flux"

Done

**References**

Jakobsson (2002):  https://doi.org/10.1029/2001GC000302

Meire et al. (2015): https://doi.org/10.5194/bg-12-2347-2015

DeGrandpre et al. (2020): https://doi.org/10.1029/2020GL088051

**Reviewer 2**

Review of: High interannual surface pCO2 variability in the Southern Canadian Arctic Archipelago's Kitikmeot Sea by Sims et al

Overview: The MS by Sims et al presents new data for the Canadian Arctic Archipelago in the area of the Kitikmeot Sea. The observations are largely collected from a ship-of-opportunity,

but also data from a seafloor observatory platform and an eddy covariance flux tower are also presented. Four years of summer observations are used to define this region as a sink for atmospheric CO2. Notable spatial differences in the data were highlighted, as were differences from the seafloor platform and the flux tower. My impression with this study is that the data are unique but I wonder about the discussions around the comparison to the flux tower and the seafloor node, and I can't help but think about the missed opportunity to compare these new results to data in the SOCAT holdings. For instance, a quick check of SOCAT reveals there are underway surface measurements in this area from Mike DeGrandpre for the years of 2017, 2019, and 2020, not to mention the earlier data from Tim Papakyriakou. It isn't clear to me that the ONC data is directly comparable to the surface data given they are measurements from the sea floor, and without understanding the size of the "footprint" of the EC tower-determined surface pCO2, that isn't a directly obvious comparison point either. Perhaps linking to the SOCAT holdings could benefit various sections of the discussion, as well as lead to a discussion about trends beyond inter-annual variability. Discussion of the drivers of the spatial and temporal variability is pretty limited and could be strengthened as well. There are also some corrections that need to be made in the presentation of the methods. I urge the authors to consider these points in addition to my detailed comments below. This paper certainly is worthy of publication in Ocean Science after the authors address these comments, and I really liked seeing the setup on the R/V Martin Bergmann. Best of luck, Wiley Evans.

We thank the reviewer for their kind and constructive feedback. The ONC mooring and EC tower make up a large part of our discussion; we recognise that we have not provided all the necessary ancillary information about these observatories in the manuscript. We have now clarified the footprint size and the uncertainties going into the calculation of $pCO_2$ from the tower. We have also provided more information about the representativeness of the ONC mooring and the impact of the freshwater plume. The biggest request from the reviewer is that we include SOCAT data, as we detail below Ahmed et.al 2019 provide the majority of the SOCAT data for the region (5 of the 7 cruises), we use their curves as these were what was used in the Ahmed et.al 2019b flux estimate. We have clarified our choice of wording in the methods to make it clear we followed the Dickson et.al 2007 SOP. Reviewer 1 also highlight that the processes driving variability could be improved. We have taken this on board and calculate the change in pCO2 due to warming using the Takahashi et.al 1993 equation as in Ahmed et.al 2019, the impact of ice melt as in Meire et.al 2015 and the impact of air sea exchange and NCP as in DeGrandpre et.al 2020.

Specific comments:

ONC seafloor platform depth is reported to be 7 and 9 m on pages 3 and 19, respectively? Which is correct? Seems like a potentially big difference in the stratified Arctic.

The mooring is at a depth of 9 m but as the mooring is big the sensor depth is 7 m, for clarity only the 7 m sensor depth is discussed in the paper.

Page 7, please report the scale used to present salinity observations.

Practical salinity was measured, which is of course unitless.

Line 185, page 9, the need for water vapor correction stems from the fact that drying removes water vapor and this impacts the partial pressure. For instance, unadjusted pCO2 from a GO8050 (that has drying components) would a "dry air" value that needs to be "corrected" to 100% humidity using SST and salinity to compute vapor pressure and adjust pCO2 to a "wet air" value. If an analytical system does not dry, then there is no need for a water vapor correction. Therefor this statement and the application of a water vapor correction to the data is in error.

You are correct that the water vapour correction is used to correct for drying. However as the Licor measures water vapour, if the water vapour measured in the Licor is not at 100% humidity for whatever reason, then this correction to 100% humidity in the equilibrator is still needed. Additionally the correction is additive so where humidity is already 100% the correction to 100% humidity makes no difference at all to $pCO_2$. You could argue that doing a correction here is superfluous but this is not an error in the approach.

**See note at top of this document. This change has been made as the reviewer suggested.**

Line 199, page 9, the LI-840 does not measure pCO2. The measurement is CO2 absorption that is linearized to produce CO2 mole fractions over a broad range. See: https://www.licor.com/env/support/LI-840A/topics/theory.html. Raw xCO2 from the instrument would be calibrated using multiple reference gases, and this calibration function should be a linear fit between the reference gas concentrations and the raw xCO2– not a piece-wise linear fit. The first point here is an easy correction to the text, the second point needs addressing at the data processing level.

You are correct that the Licor analyses $xCO_2$ not $pCO_2$.The analyser output is given in $pCO_2$ which was converted back to $xCO_2$ for calibrations. I can see that this has caused confusion; "measured" has been replaced with "the output provided" for clarity here.

**See note at top of this document. This change has been made as the reviewer suggested.**

Page 9, the authors state in situ temperature and salinity were adjusted for "ubiquitous" skin effects when calculating "interfacial" pCO2. I believe this means pCO2@equilT was adjusted to pCO2@skinT, but is presented as pCO2@SW. Was the relationship from Takahashi et al also used for the salinity adjustment?

We do not actually measure $pCO_2$@skin so are careful not to present the data that way. We assume that the temperature and salinity skin effects impact $pCO_2$ and adjust them to what we expect $pCO_2$ to be at the skin. Skin temperature and skin salinity values were used for all equations needing seawater temperature and salinity as recommended by Woolf et.al 2016. Woolf et.al 2016 say that "Unfortunately, the correction for salinity is quite uncertain since it will vary according to the circumstances of the salinity change", Takahashi et.al 1993 give a value of 0.94 and Sarmiento and Gruber et.al 2006 give a value of 1, given the associated uncertainty with this correction it was decided that this should not be included.

Page 9, The Wanninkhof 2014 relationship is used for Schmidt number but Nightingale et al 2000 was used for the gas transfer rate, why is that? Why not use Wanninkhof 2014? Also, was there good agreement between the reanalysis and locally observed wind speeds?

Nightingale et al 2000 do not provide schmidt numbers, the Wanninkhof et.al 2014 schmidt numbers are the best available. Wind speed differences between the reanalysis and the locally observed winds were not compared. Any small differences in the wind are unlikely to change the magnitude of the flux as there was a large $\Delta pCO_2$ most of the time.

Page 10, expressing pCO2 uncertainty as an absolute value is a bit misleading as certainly the uncertainty is less than 8 uatm at 200 uatm and likely more than 8 at 600 uatm. Instead of "final" could say "average"? Suggest sticking to expressing uncertainty as a percentage.

This is a good point and final has been changed to average. We have used the +/- 5uatm from DeGrandpre et al. (2020) which is an absolute value not a percentage. It is difficult to combine this with the uncertainty coming from the temperature correction which we express as a %. The reviewer correctly identifies that we have had to use a pCO2 value of 300 uatm to get a final/average value in uatm.

Page 10, somewhere the size of the footprint of the EC tower needs to be defined. Also, what are the uncertainties in SW pCO2 determined for the EC tower?

This is a very fair request and something that should be in the text or citeable. The footprint size is now provided and the uncertainty on the tower derived pCO2 is given in the methods.

For instance, the authors use temperature and salinity from 13 m (i.e. not surface and deeper than the ONC platform) to compute Schmidt number and CO2 solubility. Does this, in addition to the spatially integrative nature of the EC tower determined SW pCO2, contribute to the reported differences from the underway pCO2 measurements. That is in addition to the surface skin effects? Given underway pCO2 was adjusted for median surface skin effects.

The following has been added to the text "The large differences between the methods can not be reasonably explained by changes due to SST. Accounting for a thermal skin of 0.17°C in the RV Bergmann data only alters the $pCO_{2\ (sw)}$ by about ~ 6 µatm based on the 4.23% °C$^{-1}$ Takahashi et al. (1993) constant. Similarly for the *RV Martin Bergmann* to match values from the EC tower the SST at the surface would need to be ~5 °C greater than measured by *RV Martin Bergmann* at 1 m, modelling studies suggest this is unlikely to be the case (Xu et al., 2021). If the SST from the 13 m mooring is not representative of the interface then it is possible that the schmidt number and $k_0$ may be biased, even if this were the case the magnitude of the impact can not explain the $pCO_{2\ (sw)}$ differences between the methods."

To really dig into this we would need vertical information about the $CO_2$ system and temperature and salinity. We are currently working that CTD data up into another manuscript, including that data was unfortunately beyond the scope of a single manuscript.

Page 13, Lines 281-294, 2016 doesn't look to be "close to atmospheric equilibrium" in Figure 4, though the areal averages in Table 1 indicate conditions were closer to atmospheric levels than during the other years. I was surprised by the degree of variability during 2016 relative to the other years. Maybe this is a point that could be built on RE drivers?

We agree with this, for clarity the sentence has been changed to "*There was high interannual pCO2 (sw) variability (Table 1), where average measured pCO2 (sw) was closer to equilibrium with the atmosphere (404 µatm) in 2016 than in 2017 (309 µatm), 2018 (261 µatm) and 2019 (331 µatm) where it was highly undersaturated (Figure 4d).*".

Table 1 legend: the cautionary note seems a bit odd since the comparison between years is done in Discussion section 4.3. Maybe remove this statement?

Done

Figure 5 and section 4.1: the legend states "surface pCO2 from across the Kitikmeot Sea" which I think means all the data in Figure 4. But it doesn't look like all the data are shown. Suggest to use only data from within the EC tower footprint, whatever that is, so as to be more directly comparable. This might help clarify this section and better support the statement on Lines 330-332. The additional SOCAT data might also help in this section as well.

I can confirm that all the data from figure 4 are shown on figure 5; it may appear this way because of the longer temporal sampling period of the tower and the mooring. There was not a lot of direct overlap between the Bergmann and tower; this is why it is stated in the text every time there was overlap and what the values were.

I don't' follow the comparison to the seafloor platform without some understanding of how temperature and salinity also compare. Could this be added to Figure 5?

The following text has been added in the introduction. "(Duke et al., 2021) confirmed that the biogeochemical measurements at the ONC site were representative of the offshore during most seasons by comparing discrete DIC and TA samples collected at both 2 and 7 m at the ONC platform and (station) B1. Additionally, following the sea ice melt and river runoff period (2 weeks in early July), the surface stratification at ONC brakes down, and the DIC, TA, salinity, and temperature values measured then again become representative of the surface mixed layer." Temperature and salinity from the ONC platform are presented in (Duke et al., 2021).

Sections 4.2 and 4.3 would benefit from comparison with the SOCAT data holdings.

SOCAT v2021 were checked for relevant data during the analysis of this data. At the time there was only one cruise of data (2005 measurements from Odin provided by Fransson, A) for the region besides what is presented in Ahmed et.al 2019. The Mike DeGrandpre data were only added to SOCAT v2022 which was released on June 14[th], 6 weeks before the submission of this manuscript. Additionally, most of the Mike DeGrandpre was collected in the Beaufort Sea and the Amundsen Gulf, only one Mike DeGrandpre cruise does a full passage through the Kitikmeot Sea, as that was in 2020 we do not have our own data to compare against. The 2010 to 2016 Amundsen cruises provide the vast majority of the SOCAT data for the Kitikmeot Sea, even in SOCAT v2022 they account for 5 of the 7 cruises. Ahmed et.al 2019 already nicely present and interpret that data, the Ahmed et.al 2019

curves are used as they synthesise the existing cruise data for the whole Archipelago and are what are used for the estimation of the regional flux in Ahmed et.al 2019b.

Table from SOCAT data viewer.

| expocode | platform_name | platform_type | investigators | qc_flag | socat_version | documentation | download | crossovers | qc flags |
|---|---|---|---|---|---|---|---|---|---|
| 18DL2010 0701 | CCGS Amundsen | Ship | "Papakyri akou, T." | D | 2019.0 N | Docume ntation | Save As... | Check for crossovers | Examine QC Flags |
| 18DL2011 0718 | CCGS Amundsen | Ship | "Papakyri akou, T." | D | 2019.0 N | Docume ntation | Save As... | Check for crossovers | Examine QC Flags |
| 18DL2014 0707 | CCGS Amundsen | Ship | "Papakyri akou, T." | D | 2019.0 N | Docume ntation | Save As... | Check for crossovers | Examine QC Flags |
| 18DL2015 0417 | CCGS Amundsen | Ship | "Papakyri akou, T." | D | 2019.0 N | Docume ntation | Save As... | Check for crossovers | Examine QC Flags |
| 18DL2016 0802 | CCGS Amundsen | Ship | "Papakyri akou, T." | D | 2019.0 N | Docume ntation | Save As... | Check for crossovers | Examine QC Flags |
| 18SN2020 0907 | Louis S. St-Laurent | Ship | "DeGrand pre, M." | D | 2022.0 N | Docume ntation | Save As... | Check for crossovers | Examine QC Flags |
| 77DN2005 0720 | Oden | Ship | "Fransson , A." | D | 3.0N | Docume ntation | Save As... | Check for crossovers | Examine QC Flags |

Section 4.4 title, suggest replace "sink for pCO₂" with "sink for atmospheric CO₂"

Done

Data availability: while I appreciate that the authors are making their data available through Zenodo, and I applaud them for the effort, these data would be a bigger benefit to the community if they are submitted to SOCAT and NCEI. I strongly suggest the authors consider submitting these data to SOCAT.

We are committed to making these data available and have always planned to submit the data to SOCAT after peer review.